# Adversarial Attack and Defense for Denoising Diffusion Sampling

Zhao-Rong Lai [1]  Xiwen Yuan [2]  Jian Weng [3]

## Abstract

Denoising diffusion sampling (DDS) is an emerging approach for generating new samples that have the same distribution as some training samples. However, it is vulnerable to adversarial attacks by even a Gaussian perturbation. In this work, we propose a complete set of adversarial attack and defense methodology for DDS. In the attack side, we propose to inject a perturbation to the sampling stage, which significantly worsen the performance of sample generation. In the defense side, we propose a local variation based regularization model for the potential function minimization, which effectively tolerates the adversarial perturbations. Moreover, we develop a conjugate gradient algorithm to solve the defense model, which integrates with a recently-developed zeroth order rejection sampling method that saves computational cost. Experimental results show that the proposed attack significantly worsen the existing state-of-the-art methods, but can be defended by the proposed local variation regularization.

## 1. Introduction

Sampling from a distribution is a fundamental task in machine learning. It is widely used not only in traditional areas like decision making, statistical inference, and optimization, but also in frontier areas like diffusion generative modeling, molecular dynamics, and differential privacy. Due to recent developments of diffusion models, it is now possible to generate new samples that have the same distribution as some training samples, instead of from the exact distribution itself. This significantly improves the convenience of

[1]School of Mathematics and Information Science, Guangzhou University, Guangzhou 510006, China [2]Department of Mathematics, College of Information Science and Technology, Jinan University, Guangzhou, China [3]Cyberspace Institute of Advanced Technology, Guangzhou University, Guangzhou 510006, China. Correspondence to: Xiwen Yuan <yxw20041225@stu2023.jnu.edu.cn>.

*Proceedings of the 43rd International Conference on Machine Learning*, Seoul, South Korea. PMLR 306, 2026. Copyright 2026 by the author(s).

sampling under certain assumptions. For example, some diffusion processes and gradient flows require an isoperimetric property (Vempala & Wibisono, 2019; He et al., 2020; Salim et al., 2022; Carrillo & Skrzeczkowski, 2025). Some diffusion generative models require an approximation of the score function with certain accuracy (Benton et al., 2024; Chen et al., 2023a;b; Li et al., 2024). Some Monte Carlo (MC) based methods (He et al., 2024) require the oracle minimum of the potential function $V(x)$ (i.e., the unnormalized density of $x \propto \exp(-V(x))$). These methods are able to handle various sampling tasks on unnormalized, log-concave, or non-log-concave densities, etc.

Generally speaking, the mechanism of the above-mentioned sampling approach is to add a Brownian noise $B_t$ to a training sample $X_t$ in the forward process, and then denoise with another Brownian noise $\bar{B}_t$ (independent of $B_t$) in the backward process. By this means, a new sample $\bar{X}_t$ that is different from $X_t$ but has the same distribution as $X_t$ can be generated. A crucial stage of this approach is the Denoising Diffusion Sampling (DDS), which aims to recover the underlying distribution and generate a new sample. However, it is widely-recognized that sampling can be vulnerable to adversarial attacks (Lee et al., 2018; Ben-Eliezer & Yogev, 2020; Braverman et al., 2021; Ben-Eliezer et al., 2022; Woodruff & Zhou, 2024). In a decentralized setting (Lian et al., 2017; Koloskova et al., 2020; Cyffers & Bellet, 2022; Cyffers et al., 2023), the variable value at the previous timestamp $x_{t_{k-1}}$ should be transmitted from the previous user $(k-1)$ to the next user $k$. During this transmission, another perturbation $\delta_t$ can be mixed into the current variable value $x_{t_k}$ intentionally or unintentionally. Besides, some accidental or intrinsic factors may also cause such a perturbation, like channel interference and sampling errors. Worse still, such an adversarial attack can happen at each of the hundreds or even thousands of timestamps in an entire DDS process. Note that this perturbation $\delta_t$ is different from the above-mentioned Brownian noises $B_t$ and $\bar{B}_t$. Instead, $\delta_t$ aims to damage the adding of $B_t$ and the denoising with $\bar{B}_t$, in order to sabotage the generation of new correct samples. Even a prevalent Gaussian perturbation can significantly diminish the performance of DDS, as shown in Figures A2(a)(b), A3(a)(b), A4, A5, A8(a)(b), A9, A10, A13(a)(b), A14, A15, and A18(a)(b). Hence this vulnerability is rather significant.

To develop an adversarial defense scheme, we notice that the methodology of total variation (TV) is a traditional denoising approach that has been widely used in signal processing, image restoration, optimal control, and data transmission (Rudin et al., 1992; Chan et al., 2006; Chen et al., 2006). Recently, it has been verified to be effective in invariant feature extraction and out-of-distribution generalization (Lai & Wang, 2024; Wang et al., 2025). Therefore, we extract a local segment $|\nabla V(x)|$ of TV, to quantify the **local variation (LV)** of potential function $V(x)$ with respect to (w.r.t.) the variable value $x$. LV can effectively capture the adversarial perturbations inside the DDS process. On the other hand, DDS mostly depends on MC methods with thousands of intermediate MC samples $z$, which are mostly first-order methods (i.e., requiring $\nabla V(z)$), such as the Reverse Diffusion Monte Carlo (RDMC, Huang et al. 2024a), the Recursive Score Diffusion-based Monte Carlo (RSDMC, Huang et al. 2024b), and the stochastic localization via iterative posterior sampling (SLIPS, Grenioux et al. 2024). We develop a conjugate gradient algorithm to solve the proposed LV regularization model, which integrates with a recent MC method requiring only $V(z)$, called Zeroth Order Diffusion-Monte Carlo (ZOD-MC, He et al. 2024). This approach saves computational costs by lowering the order of queries. Our main contributions can be summarized as follows, while the diagram of our method is shown in Figure 1.

**1.** We propose an adversarial attack approach on DDS, which injects a perturbation into the variable value during its computation and transmission.

**2.** We propose a minimization model of the LV regularized potential function, which can tolerate the adversarial perturbations injected in the variable value. It forms an Adversarial Defense method for DDS (**ADDDS**).

**3.** We develop a conjugate gradient algorithm to solve the proposed ADDDS model, which has theoretical guarantees on convergence. It integrates with the zeroth-order method ZOD-MC, which saves computational costs.

## 2. Preliminaries and Related Works

We introduce some preliminaries and related works on DDS, adversarial attacks, and TV.

### 2.1. Denoising Diffusion Sampling

As introduced in Section 1, a diffusion model in this field aims to generate a new sample $\bar{X}_t$ that subjects to the same distribution $p$ as that of a training sample $X_t$. Suppose the dimensionality of $X_t$ is $d$. A common methodology is to exploit the Ornstein-Uhlenbeck (OU) process:

$$\mathrm{d}X_t = -X_t\,\mathrm{d}t + \sqrt{2}\,\mathrm{d}B_t, \quad X_0 \sim p, \qquad (1)$$

where $B_t$ is the Brownian noise for the forward process that constructs the training sample $X_t$. $X_t$ can be decomposed into two components: $X_t = e^{-t}X_0 + \sqrt{1 - e^{-2t}}Z$, where $(X_0, Z) \sim p \otimes \gamma^d$ and $\gamma^d$ denotes the $d$-dimensional standard Gaussian distribution. We denote the distribution of $X_t$ by $p_t := \mathrm{Law}(X_t), \forall t \geqslant 0$. When the terminal time $T$ of process (1) is large enough, $p_T$ is close to $\gamma^d$. Then the backward process $\{\bar{X}_t\}_{0 \leqslant t \leqslant T}$ can be constructed by reversing the OU process (1) from time $T$ to 0, which means that $\mathrm{Law}(\bar{X}_t) = \mathrm{Law}(X_{T-t})$ for all $t \in [0, T]$. This leads to the following denoising diffusion process:

$$\mathrm{d}\bar{X}_t = (\bar{X}_t + 2\nabla \log p_{T-t}(\bar{X}_t))\,\mathrm{d}t + \sqrt{2}\,\mathrm{d}\bar{B}_t,$$
$$\bar{X}_0 \sim \gamma^d, \quad 0 \leqslant t \leqslant T, \qquad (2)$$

where $\bar{B}_t$ is another Brownian noise and independent of $B_t$. $\nabla \log p_t$ is called the *score function* for $p_t$, which is not explicitly known. A widely-used method is to learn the score function by denoising score matching methods with MC (Ho et al., 2020; Sohl-Dickstein et al., 2015; Song et al., 2021):

$$s(t, x) := \frac{1}{n(t)} \sum_{i=1}^{n(t)} \frac{e^{-t}z_{t,i} - x}{1 - e^{-2t}}, \qquad (3)$$

where $\{z_{t,i}\}_{i=1}^{n(t)}$ are the independent samples from MC at time $t$. If the learned score $s(t, x)$ satisfies the following error bound

$$\mathbb{E}_{x \sim p_t}[\|s(t, x) - \nabla \log p_t(x)\|_2^2] \leqslant \varepsilon_{score}^2, \ \forall\, 0 \leqslant t \leqslant T, \quad (4)$$

non-asymptotic convergence for diffusion model (2) can be guaranteed in theory (Benton et al., 2024; Chen et al., 2023a;b). $\varepsilon_{score} \geqslant 0$ denotes a preset error tolerance. In practice, a common method to simulate (2) is to use discretizations like the Euler Maruyama or an exponential integrator. This can be vulnerable to adversarial attacks or sampling errors, which will be illustrated in Section 3.1. A recent method ZOD-MC develops a rejection sampling scheme that requires only a zeroth-order query to the potential value $V(z_{t,i})$ at each $z_{t,i}$ (He et al., 2024). The crucial stage of ZOD-MC is to solve for the minimum of $V$, which just needs to be implemented for only once for all the MC samples $\{z_{t,i}\}$ during the entire DDS process:

$$\min_{x \in \mathbb{R}^d} V(x). \qquad (5)$$

This is also the crucial part that we can exploit to develop an adversarial defense strategy, as illustrated in Section 3.2.

### 2.2. Adversarial Attacks

An adversarial attack deliberately injects a small perturbation, which can be hardly recognized, into the given data, in order to make a learning system give incorrect or distorted results. While the training of a learning system aims to minimize a given loss function, the adversarial attack aims to maximize this loss function with a perturbed input $(x + \delta)$:

$$\max_{\|\delta\|_{\mathfrak{p}} \leqslant \epsilon} \mathcal{L}(x + \delta), \qquad (6)$$

where $\|\delta\|_{\mathfrak{p}} \leqslant \epsilon$ means that the $\ell_{\mathfrak{p}}$ norm of the perturbation $\delta$ should be no larger than a preset *budget* $\epsilon \geqslant 0$. $\mathcal{L}$ is the loss function for training, which considers both unsupervised

*Figure 1.* Diagram of the entire adversarial attack and defense scheme for denoising diffusion sampling in a decentralized scenario. While variable $x_k$ (the simplified version of $x_{t_k}$) is transmitted from the $(k-1)$-th user to the $k$-th user, an adversarial perturbation $\delta_{k-1}$ can be injected to yield a perturbed variable $\tilde{x}_k$. In the defense side, the local variation regularized $V^{\bullet}$ not only tolerates adversarial perturbations, but also integrates with a zeroth-order rejection sampling scheme to produce MC samples $\{\tilde{z}_{t,i}\}$. Then the score estimation and backward process can be implemented to produce the next variable $x_{k+1}$. In the next round, $x_{k+1}$ may still be attacked and the same attack-defense dynamic continues.

learning and supervised learning (the dependent variable $y$ can be embedded in $\mathcal{L}$ and thus omitted in this setting).

A fundamental attack approach is the Fast Gradient Sign Method (FGSM, Goodfellow et al. 2015), which injects a small perturbation in the same direction of the gradient sign:

$$\tilde{x} = x + \epsilon' \cdot \text{sign}(\nabla_x \mathcal{L}(x)), \qquad (7)$$

where $0 \leqslant \epsilon' \leqslant \epsilon / \|\text{sign}(\nabla_x \mathcal{L}(x))\|_{\mathfrak{p}}$.

Besides FGSM, the Projected Gradient Descent (PGD, Madry et al. 2018) is a multi-iteration attack strategy that aims to achieve a stronger attack of "first-order adversary":

$$\tilde{x}_{(l+1)} = \text{proj}_{\mathcal{B}_{\mathfrak{p}}[x;\epsilon]}(\tilde{x}_{(l)} + \eta \cdot \text{sign}(\nabla_x \mathcal{L}(x))), \quad (8)$$

where $\eta > 0$ denotes the step size for a single PGD iteration, and $\text{proj}_{\mathcal{B}_{\mathfrak{p}}[x;\epsilon]}$ denotes the projection operator onto the closed neighborhood of $x$ with a radius of $\epsilon$. A common defense approach for adversarial attack is the adversarial training (Briglia et al., 2025; Zhang et al., 2021), which takes the adversarial samples into an improved training scheme. However, it is a training-time defense that modifies the model parameters, which is different from the runtime defense ADDDS that operates during the inference/sampling process to purify the perturbed trajectory.

### 2.3. Total Variation

TV (Rudin et al., 1992) is a traditional instrument that aggregates the variation of a function $f(x)$ while the variable $x$ traverses the domain $\Omega$. It is widely-recognized that many kinds of noise and other unwanted fine-scale details can be captured by TV. Suppose $\Omega \subseteq \mathbb{R}^d$ is an open set and let $L^1(\Omega)$ be the corresponding Lebesgue-integrable function space. Then the TV of $f \in L^1(\Omega)$ can be defined as (Chan et al., 2006):

$$\int_{\Omega} |\nabla f| := \sup\{ \int_{\Omega} f(x)\text{div}g(x)\, dx :$$
$$g \in C_c^1(\Omega, \mathbb{R}^d), \|g\|_{L^{\infty}(\Omega)} \leqslant 1\}, \qquad (9)$$

where $g$ works as a probe function with its divergence $\text{div}g$, which has a compact support contained in $\Omega$ and an essential supremum no larger than 1. TV calculates the local

variations (LV) of $f$ across all dimensions of $x$, and then accumulates these LVs over the entire domain $\Omega$. $|\nabla f|$ is symbolic if $f$ is non-differentiable, but it is exactly the $\ell_2$ **norm of the gradient** $\nabla f$ **if** $f$ **is differentiable**. In practice, the literature usually considers the functions with bounded variation (BV) $\{f \in L^1(\Omega) : \int_{\Omega} |\nabla f| < \infty\}$.

Based on the essence of TV, the TV-$\ell_1$ model (Rudin et al., 1992) is proposed to recover the main component of $f$ and reduce noise and other unwanted fine-scale details simultaneously:

$$\inf_{\hat{f} \in L^2(\Omega)} \left\{ \int_{\Omega} |\nabla \hat{f}| + \lambda \int_{\Omega} (f - \hat{f})^2\, dx \right\}, \qquad (10)$$

where the second term measures the approximation of the recovery $\hat{f} \in L^2(\Omega)$ to the target function $f$, and $\lambda \geqslant 0$ is the approximation hyperparameter. The first term can be considered as a regularization term that tolerates the TV of $\hat{f}$.

## 3. Methodology

DDS is an effective method for new sample generation under an ideal and pure environment where interference and perturbations are absent. However, it can be vulnerable to adversarial attacks or sampling errors. In this section, we propose a complete set of adversarial attack and defense methodology for DDS, which is briefly demonstrated in Figure 1.

### 3.1. Attack Side for Denoising Diffusion Sampling

A vulnerable stage of DDS lies in the following update via an exponential integrator:

generate $\xi_k \sim \gamma^d$ such that $\xi_k$ is independent of $\xi_0, \cdots, \xi_{k-1}$,

$$x_{k+1} \leftarrow e^{t_{k+1} - t_k} x_k + 2(e^{t_{k+1} - t_k} - 1)s(T - t_k, x_k)$$
$$+ \sqrt{e^{2(t_{k+1} - t_k)} - 1}\xi_k, \qquad (11)$$

**where** $x_{k+1}$ **is a simplified notation of** $x_{t_{k+1}}$, which is the variable value at the $(k+1)$-th timestamp. $s(T - t_k, x_k)$ is the estimated score function value at time $(T - t_k)$ and variable value $x_k$, which corresponds to the term

$\nabla \log p_{T-t_k}(\bar{X}_{t_k})$ in (2). The generated $\xi_k$ serves as the Brownian motion from $t_k$ to $t_{k+1}$. In a decentralized setting (Lian et al., 2017; Koloskova et al., 2020; Cyffers & Bellet, 2022; Cyffers et al., 2023), $x_k$ should be transmitted from the $(k-1)$-th user to the $k$-th user, which is exposed to intentional or unintentional perturbations. Specifically, $x_k$ can be perturbed due to channel interference, which consequently affects the score estimation $s(T - t_k, x_k)$. Besides, $\xi_k$ may involve a sampling error. All these perturbations can be summarized as the following update without loss of generality:

$$x_{k+1} \leftarrow e^{t_{k+1}-t_k} x_k + 2(e^{t_{k+1}-t_k} - 1)s(T - t_k, x_k)$$
$$+ \sqrt{e^{2(t_{k+1}-t_k)} - 1}\xi_k + \underline{\delta_k}, \tag{12}$$

**where $\delta_k$ represents the overall perturbation in this update.** As explained in Section 2.2, if $\|\delta_k\|_{\mathfrak{p}}$ is small enough, such a perturbation can be hardly recognized but can sabotage the generation of $x_{k+1}$. Worse still, this adversarial attack can happen at each of the hundreds or even thousands of timestamps $\{t_k\}_{k=0}^N$ in an entire DDS process, leading to the bad performance in Section 4 and Appendix B.

**As for the construction of $\delta_k$, we find that even a Gaussian perturbation with a small standard deviation (STD) can significantly diminish the performance of DDS. Since Gaussian perturbations are prevalent in various real-world applications, this vulnerability is rather significant.** Besides, FGSM and PGD in (7) and (8) can also be adopted to construct $\delta_k$. To do this, we design the following loss function:

$$\mathcal{L}_{T-t_k}(x) := \frac{1}{n(T-t_k)} \sum_{i=1}^{n(T-t_k)} \left\| \frac{e^{-(T-t_k)} z_{T-t_k,i} - x}{1 - e^{-2(T-t_k)}} \right\|_2^2. \tag{13}$$

It averages the squared $\ell_2$ norms of the components of the score function at time $(T - t_k)$. By this means, the constructed perturbation $\delta_k$ aims to **maximize the score function divergence**, in order to sabotage the second term in (12). Therefore, the proposed attack is fundamentally **diffusion-specific** and goes beyond a general perturbation noise model, because it exploits the vulnerability of the denoising diffusion process and its reliance on **score matching**.

### 3.2. Defense Side for Denoising Diffusion Sampling

The defense side for DDS aims to tolerate the adversarial perturbations $\{\delta_k\}_{k=0}^N$ by all means. We find that the crucial technique of defense for DDS lies in the computation of the potential function $V(x)$. Since the variable $\tilde{x} = x + \delta$ has been attacked by a perturbation $\delta$, the computation of $V(\tilde{x})$ should also consider this interference. As introduced in (5), the crucial stage of the zeroth order MC method ZOD-MC is

to minimize $V(x)$. Therefore, in the presence of adversarial perturbations, we propose to minimize $(V(x) + \lambda|\nabla V(x)|)$:

$$\min_{x \in \mathbb{R}^d} \{V(x) + \lambda|\nabla V(x)|\}, \tag{14}$$

where $\lambda|\nabla V(x)|$ is an LV regularization term with strength $\lambda \geqslant 0$. First, we provide a theoretical result on the equivalence between (14) and (5), which enables the rejection sampling scheme in (He et al., 2024).

**Theorem 3.1.** *If $V(x)$ is differentiable, then (14) and (5) have the same solution set.*

The proof is provided in Appendix A.1. There are generally no closed-form solutions to both (14) and (5), thus iterative algorithms should be developed to solve them. Although they have the same solution set, their iteration paths are substantially different. The iteration path of (14) simultaneously minimizes $V(x)$ and tolerates the perturbation of $V$ induced by that of $x$, but the iteration path of (5) only minimizes $V(x)$. Since $V(x)$ is generally a nonconvex function, probably with complicated geometrical structures, a standard gradient-type algorithm can only find a critical point $x^{\bullet}$ of $V$. Due to the different iteration paths between (14) and (5), they also lead to different critical points. This can be demonstrated by the matched oracle complexity experiments (see the experimental settings in Appendix B.1) in Table 1. While the critical points of (5) satisfy $|\nabla V(x_{org}^{\bullet})| = 0$, the critical points of (14) yield nonzero LVs $|\nabla V(x_{ADDDS}^{\bullet})| > 0$. This is because the critical points of (14) are subject to

$$\nabla V(x_{ADDDS}^{\bullet}) \in -\lambda \cdot \partial|\nabla V(x_{ADDDS}^{\bullet})|,$$
$$\partial|\nabla V(x_{ADDDS}^{\bullet})| \text{ defined by Definition 3.2}, \tag{15}$$

which may lead to $|\nabla V(x_{ADDDS}^{\bullet})| > 0$. This mechanism allows (14) to tolerate the small LV induced by the perturbation $\delta$. Although it is obvious that $|\nabla V(x^*)| = 0$ with the ground-truth minimizer $x^*$, the perturbed version should probably be $|\nabla V(x^* + \delta)| > 0$. This situation can be adapted to by $|\nabla V(x_{ADDDS}^{\bullet})| > 0 = |\nabla V(x_{org}^{\bullet})|$. The same reason applies to why $V(x_{ADDDS}^{\bullet}) > V(x_{org}^{\bullet})$, which adapts to the situation $V(x^* + \delta) > V(x^*)$.

On the other hand, the LV model (14) is substantially different from the TV model (10) (Rudin et al., 1992; Chen et al., 2006; Chan et al., 2006; Lai & Wang, 2024; Wang et al., 2025; Lai et al., 2026). The former is a local model that aims to retrieve an optimal argument $x^*$, while the latter is a global model that aims to retrieve an entire optimal function $f^*$ in a functional space (e.g., $L^2(\Omega)$). Therefore, the existing solving algorithms (Yang et al., 2009; Barbero & Sra, 2011; Bai et al., 2016) for TV models cannot be directly applied to the LV model (14), and we need to develop a solving algorithm. Since $|\nabla V(x)|$ is non-differentiable, we need to use the Fréchet subdifferential $\partial f(x)$.

*Table 1.* Critical points $x^\bullet$, potentials $V(x^\bullet)$, and LVs $|\nabla V(x^\bullet)|$ solved from (5) and (14). While (5) yields zero LVs, (14) yields nonzero LVs that can tolerate the perturbations induced by the attacked variables $\tilde{x}$.

| Oracle Complexity | Model | Critical Point $x^\bullet$ | $V(x^\bullet)$ | $\lvert\nabla V(x^\bullet)\rvert$ |
|---|---|---|---|---|
| 200 | (5) | $(-5.96e-08, -2.98e-08)$ | 3.9966 | $5.9605e-08$ |
|  | (14) | $(2.68e-04, 3.16e-04)$ | 4.0512 | 0.0893 |
| 3200 | (5) | $(-1.19e-07, -5.96e-08)$ | 3.9966 | $1.1921e-07$ |
|  | (14) | $(6.68e-05, 1.34e-04)$ | 4.0512 | 0.0893 |
| 6200 | (5) | $(5.96e-08, -5.96e-08)$ | 3.9966 | $2.3842e-07$ |
|  | (14) | $(1.68e-04, 8.38e-05)$ | 4.0510 | 0.0887 |
| 9200 | (5) | $(5.96e-08, -5.96e-08)$ | 3.9966 | $2.3842e-07$ |
|  | (14) | $(-1.04e-04, -2.08e-04)$ | 4.0886 | 0.1288 |

**Definition 3.2** (The Fréchet Subdifferential).

$$\partial f(x) := \left\{ z \in \mathbb{R}^d : \liminf_{\substack{y \to x \\ y \neq x}} \frac{f(y) - f(x) - z^\top(y-x)}{\|y - x\|_2} \geqslant 0 \right\}. \tag{16}$$

In fact, $|\nabla V(x)|$ is non-differentiable if and only if $\nabla V(x) = 0$, where we can directly set the corresponding subdifferential $\partial|\nabla V(x)| = 0$ since it satisfies (16). Thus the complete form of $\partial|\nabla V(x)|$ can be directly calculated as follows, which works as an ordinary gradient:

$$\partial|\nabla V(x)| \leftarrow \begin{cases} \frac{(\nabla V(x))^\top \nabla^2 V(x)}{|\nabla V(x)|} & \text{if } \nabla V(x) \neq 0, \\ 0 & \text{if } \nabla V(x) = 0. \end{cases} \tag{17}$$

We treat the subgradient of the entire objective function

$$W(x) := V(x) + \lambda|\nabla V(x)|, \nabla W(x) = \nabla V(x) + \lambda\partial|\nabla V(x)| \tag{18}$$

as a whole, instead of treating $\nabla V(x)$ and $\lambda\partial|\nabla V(x)|$ separately. The reason is that both $\nabla V(x)$ and $\lambda\partial|\nabla V(x)|$ (based on Eq. 17) can be simultaneously calculated via the `autograd` scheme in deep learning architectures like Pytorch[1], hence it is more efficient to calculate $\nabla W(x)$ as a whole.

Next, we develop a conjugate gradient algorithm based on the Crowder-Wolfe scheme (Hestenes & Stiefel, 1952; Crowder & Wolfe, 1972; Gilbert & Nocedal, 1992) to solve (14) as follows.

$$x^{(l+1)} = x^{(l)} + \alpha^{(l)}\mathfrak{d}^{(l)}, \mathfrak{d}^{(l+1)} =$$
$$\left| \frac{(\nabla W(x^{(l+1)}) - \nabla W(x^{(l)}))^\top \nabla W(x^{(l+1)})}{(\nabla W(x^{(l+1)}) - \nabla W(x^{(l)}))^\top \mathfrak{d}^{(l)}} \right| \mathfrak{d}^{(l)}$$
$$- \nabla W(x^{(l+1)}), \tag{19}$$

where $\alpha^{(l)}$ is a step size that satisfies the Wolfe conditions

$$W(x^{(l+1)}) \leqslant W(x^{(l)}) + c_1\alpha^{(l)}(\nabla W(x^{(l)}))^\top \mathfrak{d}^{(l)},$$

---
[1] https://pytorch.org

$$(\nabla W(x^{(l+1)}))^\top \mathfrak{d}^{(l)} \geqslant c_2(\nabla W(x^{(l)}))^\top \mathfrak{d}^{(l)}. \tag{20}$$

In (20), $0 < c_1 < c_2 < 1$ are two constants, and $\mathfrak{d}^{(l)}$ represents the update direction at the $l$-th iteration. This algorithm guarantees convergence (Nocedal & Wright, 2006), which is detailed in Appendix A.2. Once a critical point $x^\bullet$ has been solved, the corresponding score function $V^\bullet := V(x^\bullet)$ can be used in the following recovering stage. In addition, ADDDS achieves the following adversarial robustness guarantee.

**Theorem 3.3** (Adversarial Robustness Guarantee). *For any $x \in \mathbb{R}^d$, if the potential function $V(x)$ is $L_{x,\epsilon}$-smooth ($L_{x,\epsilon} > 0$) in the neighborhood $\mathcal{B}_2[x; \epsilon]$, then $W(x)$ can serve as a surrogate upper bound for the adversarial loss w.r.t. $V(x)$ for any $\lambda \geqslant \epsilon$:*

$$\max_{\|\delta\|_2 \leqslant \epsilon} V(x + \delta) \leqslant W(x) + \frac{L_{x,\epsilon}}{2}\epsilon^2 \leqslant W(x) + \frac{L_{x,\epsilon}}{2}\lambda^2. \tag{21}$$

*Moreover, there exists some constant $\mathcal{C}_\mathfrak{p} > 0$ such that if $V(x)$ is $L_{x,\mathcal{C}_\mathfrak{p}\epsilon}$-smooth in the neighborhood $\mathcal{B}_2[x; \mathcal{C}_\mathfrak{p}\epsilon]$, then*

$$\max_{\|\delta\|_\mathfrak{p} \leqslant \epsilon} V(x + \delta) \leqslant W(x) + \frac{L_{x,\mathcal{C}_\mathfrak{p}\epsilon}}{2}\lambda^2. \tag{22}$$

The definition of $L_{x,\epsilon}$-smoothness can be found in Appendix A.3. In fact, to satisfy $L_{x,\epsilon}$-smoothness (or $L_{x,\mathcal{C}_\mathfrak{p}\epsilon}$-smoothness), a sufficient condition is that $V$ is twice differentiable in $\mathcal{B}_2[x; \epsilon]$ (or $\mathcal{B}_2[x; \mathcal{C}_\mathfrak{p}\epsilon]$). The proof is provided in Appendix A.3. To see the working mechanism of Theorem 3.3, we recall that a benign DDS method should minimize $V(x)$ as (5). Hence a white-box adversarial attack strategy should maximize $V(x + \delta)$ by injecting a perturbation $\delta$. Then Theorem 3.3 indicates that this adversarial loss shall not exceed $W(x)$ plus a small value. Moreover, this verifies that our defense strategy is to minimize the worst-case scenario, which formulates the following minimax robust optimization problem (details are provided in Appendix A.3):

$$\min_{x \in \mathbb{R}^d} \max_{\|\delta\|_2 \leqslant \epsilon} V(x + \delta) \leqslant \min_{x \in \mathbb{R}^d} \left\{ W(x) + \frac{L_{x,\epsilon}}{2}\epsilon^2 \right\}. \tag{23}$$

This result justifies our motivation of minimizing $W(x)$ in (14) and formally connects the proposed LV regularization $\lambda\|\nabla V(x)\|_2$ to the rigorous framework of adversarial robustness. The condition $\lambda \geqslant \epsilon$ means that the regularization strength should be no less than the perturbation budget in order to achieve the above adversarial robustness.

### 3.3. Recovering Stage for Denoising Diffusion Sampling

Our defense strategy integrates with the zeroth-order MC method ZOD-MC (He et al., 2024), which only needs to query $V(z)$ (instead of $\nabla V(z)$ or even high-order derivatives) for each MC variable $z$. This is more computationally efficient than the first-order methods because there can be hundreds of timestamps and thousands of MC variables at each timestamp in a DDS process. However, while ZOD-MC focuses on efficient sampling from a known potential function, the proposed ADDDS introduces the LV regularization to purify the potential function itself in the presence of adversarial perturbations.

Specifically, given a perturbed variable $\tilde{x} \in \mathbb{R}^d$, a sample $(\xi, u) \sim \gamma^d \otimes U[0,1]$ is generated to compute an intermediate variable $\tilde{z} := e^t\tilde{x} + \sqrt{e^{2t}-1}\xi$. If $u \leqslant e^{V^\bullet - V(\tilde{z})}$, then this $\tilde{z}$ is feasible for MC score estimation, otherwise $\tilde{z}$ will be rejected and resampled. In practice, ADDDS does not require the ground-truth global optimum $V^*$, but uses the dynamically tracked minimum $V^\bullet$ as a surrogate. To numerically guarantee that the acceptance probability $\mathbb{P}(\text{accept}) \in [0,1]$, we use a standard clamping mechanism

$$\mathbb{P}(\text{accept}) = \min\{1, \exp(V^\bullet - V(\tilde{z}))\}. \quad (24)$$

if $V(\tilde{z}) < V^\bullet$, then $\mathbb{P}(\text{accept})$ is clamped to 1, which means that this sample is deterministically accepted. We subsequently update the tracked minimum $V^\bullet \leftarrow V(\tilde{z})$, which keeps $V^\bullet$ as the current minimum. In this way, we essentially use a standard rejection sampler that accepts $\tilde{z}$ with probability $\frac{p(\tilde{z})}{M}$, where $p(\tilde{z}) = e^{-V(\tilde{z})}$ and the global bound $M = e^{-V^*}$. While this clamping mechanism may introduce a slight theoretical bias in the sampling distribution for non-adversarial settings, it provides a robustness gain in adversarial settings. It effectively prioritizes the acceptance of samples that align with the regularized and purified potential, so as to filter out the adversarial noise that would lead the sampling trajectory to a wrong direction. As indicated in Appendix C.5, this potential accuracy trade-off is consistent with the perturbation tolerance mechanism, and ADDDS achieves a significantly lower score estimation error than the non-defense ZOD-MC baseline under adversarial attack.

After generating sufficient feasible samples $\{\tilde{z}_{t,i}\}_{i=1}^{n(t)}$, the MC score estimation with the perturbed $\tilde{x}$ can be implemented by

$$\tilde{s}(t, \tilde{x}) := \frac{1}{n(t)} \sum_{i=1}^{n(t)} \frac{e^{-t}\tilde{z}_{t,i} - \tilde{x}}{1 - e^{-2t}}. \quad (25)$$

Last, the following perturbed backward process can be used to recover a new sample process $\{\tilde{x}_k\}_{k=0}^N$:

$$\tilde{x}_{k+1} \leftarrow e^{t_{k+1}-t_k}\tilde{x}_k + 2(e^{t_{k+1}-t_k}-1)\tilde{s}(T - t_k, \tilde{x}_k)$$
$$+ \sqrt{e^{2(t_{k+1}-t_k)}-1}\xi_k + \delta_k. \quad (26)$$

(26) differs from (12) in the score estimation: (26) uses $\tilde{s}$ in (25) with perturbation-adjusted $\{\tilde{z}_{t,i}\}_{i=1}^{n(t)}$, while (12) uses $s$ in (3) with non-perturbation-adjusted $\{z_{t,i}\}$. Hence (26) is more robust than (12) to adversarial attacks.

### 3.4. Extension to High-dimensional Settings

Adversarial robustness for image classification (Goodfellow et al., 2015; Madry et al., 2018) is an important topic involving a much higher dimensionality than the basic formulation by the related works in Section 2. Hence we further extend the proposed ADDDS to a high-dimensional setting for the image classification task. In this task, the sample $x$ is formulated as the parameters for a Bayesian neural network (BNN, Neal 2012; Song & Kingma 2021). Given a training data set $\mathcal{D} = \{(u_i, y_i)\}$ with $u_i$ and $y_i$ being the $i$-th image and its ground-truth label vector, the potential function can be defined by the following posterior energy function:

$$V(x) = -\log p(x|\mathcal{D}), \quad p(x|\mathcal{D}) \propto p(\mathcal{D}|x)p(x). \quad (27)$$

We can use a Gaussian prior with the standard deviation (STD) $\sigma_{prior}$, the cross-entropy classifier, and the BNN to construct a surrogate of $V(x)$. Then the corresponding score function (Song et al., 2021) is:

$$\log p(x) = -\frac{\|x\|_2^2}{2\sigma_{prior}^2}, \log p(\mathcal{D}|x) = \sum_{(u_i, y_i) \in \mathcal{D}} y_i^\top \log \hat{y}_i(u_i; x),$$
$$(28)$$

$$s(x) = \nabla \log p(x|\mathcal{D}) = \nabla \log p(\mathcal{D}|x) + \nabla \log p(x), \quad (29)$$

where $\hat{y}_i(u_i; x)$ denotes the estimated label vector (output) of the image $u_i$ through the BNN with parameter $x$. Based on the above formulation, we can use the OU process in (1), the score estimation in (3), and the DDS process with adversarial attack in (12). In the defense side, we can insert (28) into $V(x)$ of (27) and use the same ADDDS procedure with (14). The fundamental difference between the low-dimensional setting in Section 2 and our high-dimensional setting is that the former uses the existing data to generate new data, but the latter uses the existing data to generate the parameters of BNN. After the DDS process (either with or

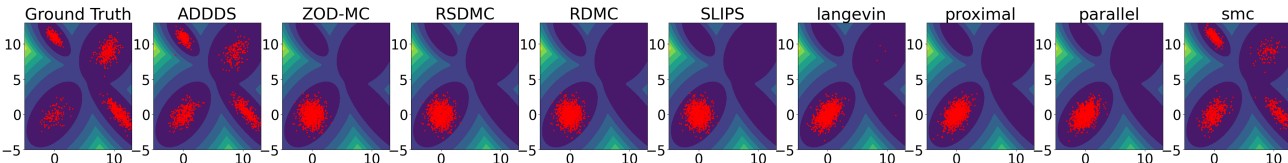

*Figure 2.* Samples generated by different methods from an asymmetric and unbalanced Gaussian mixture under $\text{PGD}^{10}$ attacks (oracle complexity= 1200).

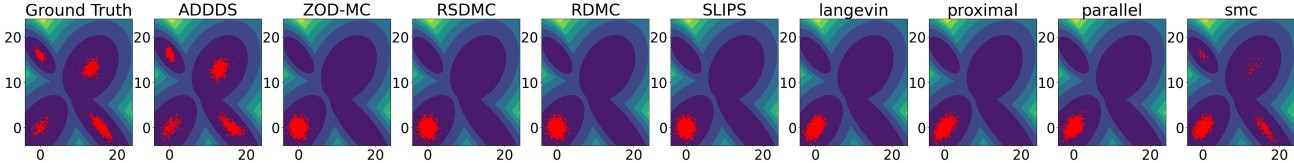

*Figure 3.* Samples generated by different methods for a Gaussian mixture with radius $R = 16$ under $\text{PGD}^{10}$ attacks.

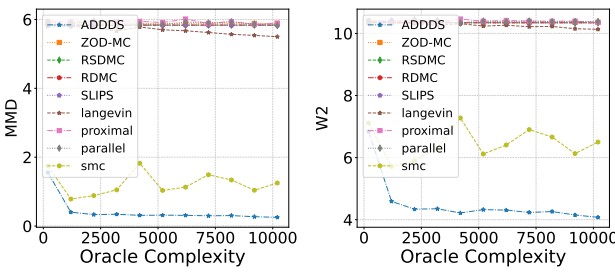

*(a)* Sampling accuracy w.r.t. oracle complexity

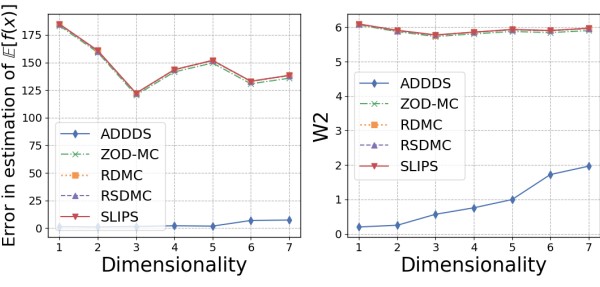

*(b)* Sampling accuracy w.r.t. dimensionality

*Figure 4.* Sampling accuracies of different methods for Gaussian mixtures under $\text{PGD}^{10}$ attacks. A lower score indicates a better performance.

without the attack or defense scheme), the generated final parameters $x_N$ can be used in BNN to perform classification tasks. Note that this high-dimensional setting is not a standard image classification training, but rather BNN posterior sampling. We implement **BNN posterior sampling** instead of classification training because it is the former that truly **conforms to the DDS setting** of this study.

## 4. Experimental Results

Our experimental setup extends the standard benchmark for DDS methods outlined in (He et al., 2024) to include more adversarial attack schemes and more baseline meth-

ods. Specifically, the plain Gaussian perturbations, FGSM (Goodfellow et al., 2015), and $\text{PGD}^{10}$ (PGD with 10-step attacks, Madry et al. 2018) are used as three attack schemes. ZOD-MC (He et al., 2024), RSDMC (Huang et al., 2024b), RDMC (Huang et al., 2024a), SLIPS (Grenioux et al., 2024), Langevin (Huang et al., 2024a;b), the proximal sampler (Liang & Chen, 2023), sequential Monte Carlo (Del Moral et al., 2006). a parallel tempering approach with MALA proposals (Lee & Shen, 2023) are used as baseline methods. The computing device has twelve Intel(R) Xeon(R) Gold 6248R CPUs, 92-GB RAM, and an Nvidia A100-PCIE-40GB GPU. **The main part of $\text{PGD}^{10}$-attack experiments are presented in the main text, while the rest experiments and detailed experimental settings are presented in Appendix B. Additional experiments including ablation studies, intrinsic defense schemes, adaptive white-box attack, computational resource requirement, acceptance rate, score estimation error, runtime comparisons, convergence of non-smooth potential function, and Hessian-based surrogate are presented in Appendix C.**

### 4.1. Gaussian Mixture Models

This experiment uses a 2D Gaussian mixture model (GMM) with unbalanced modes and non-isotropic variances, which leads to a highly asymmetrical and multi-modal problem. Model details are provided in Appendix B.1.

**Oracle Complexity and Dimensionality Dependence.** In the oracle complexity scenario, different methods are tested on different oracle complexities (i.e., total number of queries to $V(z)$ or $\nabla V(z)$). While ZOD-MC and the proposed ADDDS query $V(z)$, other baselines query $\nabla V(z)$, which gives more advantage to the latter. Figures 4(a) and A2 present the sampling accuracies w.r.t. oracle complexity for different methods. Both Maximum Mean Discrepancy (MMD) and 2-Wasserstein distance ($W_2$) are taken as evaluation metrics (the lower the better), and ADDDS outper-

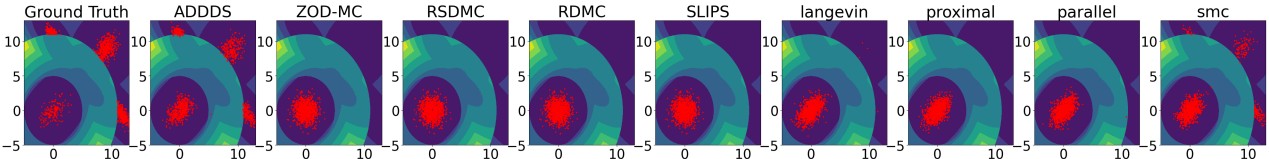

*Figure 5.* Samples generated by different methods from a discontinuous potential under PGD[10] attacks (oracle complexity= 1200).

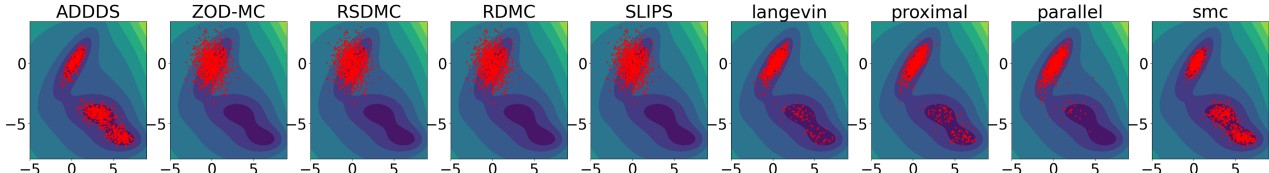

*Figure 6.* Samples generated by different methods from the Müller-Brown potential under PGD[10] attacks (oracle complexity= 1100).

forms other baselines in most cases. Figures 2, A4, A5, A6, and A7 show the generated samples of different methods with various oracle complexities, which indicate that AD-DDS recovers the best patterns on the 4-mode GMM among all the baselines across all the oracle complexities. Moreover, ADDDS requires only a few minutes of computational time (see Figure A1), which is reasonable considering its effective defense.

In the dimensionality dependence scenario, the statistics on the generated samples and $W_2$ scores of different methods w.r.t. different dimensionalities are tested, which are shown in Figures 4(b) and A3. ADDDS significantly outperforms other baselines across different dimensionalities (the plots of other baselines mutually overlap), which indicates that ADDDS scales well with dimensionality increase.

**Robustness Against Mode Separation.** In this scenario, the means of all the modes are scaled by a factor such that a mode is located at $(0, R)$, which separates different modes more distinctly and worsens the isoperimetric properties of the target distribution. Figures 7 and A8 show the results of MMD, $W_2$, and mass on center mode w.r.t. the radius $R$ for different methods. ADDDS performs more robustly than other baselines as $R$ increases, which indicates that it is effective in defending adversarial attacks under poor conditions. Moreover, generated samples in Figures 3, A9, A10, A11, and A12 show that ADDDS recovers the best patterns among all the baselines across all the radius settings.

**Discontinuous Potential.** In this scenario, a discontinuous potential function $(V(x) + U(x))$ is considered, where $U(x) = 8\lfloor\|x\|_2\rfloor\mathbb{I}_{\{5<\|x\|_2<11\}}$ and $\mathbb{I}$ denotes the indicator function that takes value 1 in the set $\{5 < \|x\|_2 < 11\}$ and value 0 otherwise. $\lfloor\cdot\rfloor$ denotes the floor operator that yields an integer. This discontinous potential strengthens the potential barrier and increases the sampling difficulty. Results in Figures 8 and A13 show that ADDDS outperforms other baselines in both MMD and $W_2$ across all the oracle

complexities. Moreover, generated samples in Figures 5, A14, A15, A16, and A17 show that ADDDS recovers the best patterns among all the baselines across all the oracle complexities.

### 4.2. Müller-Brown Potential

The Müller-Brown potential is a widely-used potential energy surface model (Müller & Brown, 1979; Sipka et al., 2023) for molecular dynamics. It is a highly nonlinear potential that has three modes, although it is the sum of four exponentials. To facilitate 2D visualization, a balanced version (Li & Tao, 2024; He et al., 2024) is used where one mode is centered near the origin. Generated samples by different methods are shown in Figures 6 and A18. Since there is no ground-truth sampling for this potential, the level curves of the potential can be used to evaluate the performance. ADDDS recovers the best patterns for the three modes among all the baselines against all the attack methods.

### 4.3. High-dimensional Experiments

We use the high-dimensional settings in Section 3.4 to conduct image classification experiments on the CIFAR 10 and CIFAR 100 data sets (Krizhevsky, 2009). The perturbation budget and the number of total sampling steps are set to $\epsilon = 0.01$ and $N = 25$, respectively. Results in Table 2 show that ADDDS outperforms all the other baselines across all the cases. Hence ADDDS is a robust defense scheme even in high-dimensional settings. Note that the absolute clean accuracies (29.81% for CIFAR 10 and 7.10% for CIFAR 100) appear low because we are not training a standard ResNet (He et al., 2016) end-to-end, but using the existing data to generate the parameters of a BNN. In this specific BNN sampling context, a clean accuracy of 29.81% for AD-DDS on CIFAR 10 is already competitive. For comparison, the standard ZOD-MC baseline only achieves 12.03%, and

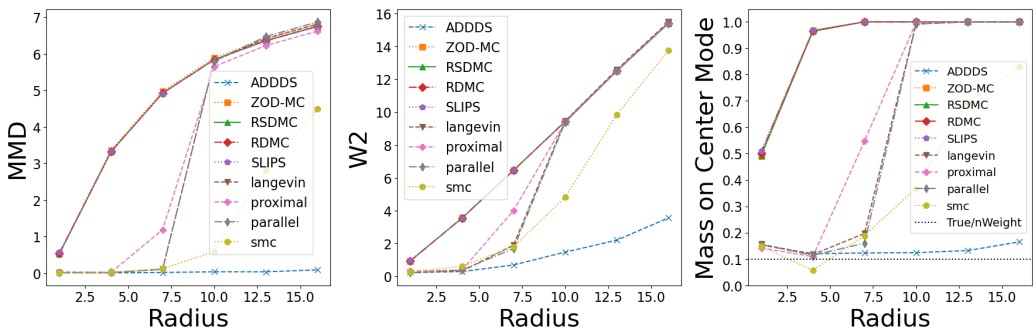

*Figure 7.* Sampling accuracies of different methods w.r.t. mode separation for Gaussian mixtures under PGD[10] attacks. A lower score indicates a better performance.

*Table 2.* Classification accuracies (%) of different methods on the CIFAR data sets under different attacks.

| Perturbation | ADDDS | ZOD-MC | RSDMC | RDMC | SLIPS | langevin | proximal | parallel | smc |
|---|---|---|---|---|---|---|---|---|---|
| | | | | | CIFAR 10 | | | | |
| Clean | **29.810** | 12.030 | 24.180 | 28.720 | 9.890 | 11.490 | 28.800 | 27.020 | 20.150 |
| Gaussian, $\sigma = 0.01$ | **29.520** | 11.820 | 23.040 | 28.460 | 9.480 | 10.200 | 28.330 | 26.210 | 19.750 |
| Gaussian, $\sigma = 0.1$ | **29.230** | 11.390 | 22.430 | 28.180 | 9.060 | 9.950 | 21.040 | 18.530 | 19.680 |
| FGSM | **14.970** | 8.830 | 5.520 | 13.920 | 8.750 | 7.370 | 7.140 | 6.630 | 13.490 |
| PGD[10] | **9.230** | 8.860 | 4.790 | 8.120 | 4.330 | 5.740 | 5.810 | 5.730 | 8.810 |
| | | | | | CIFAR 100 | | | | |
| Clean | **7.100** | 5.810 | 6.840 | 3.900 | 0.940 | 5.630 | 1.980 | 1.820 | 3.640 |
| Gaussian, $\sigma = 0.01$ | **6.940** | 1.370 | 6.660 | 3.870 | 0.830 | 1.830 | 1.150 | 1.620 | 3.400 |
| Gaussian, $\sigma = 0.1$ | **6.880** | 1.140 | 6.530 | 3.710 | 0.820 | 1.600 | 1.280 | 1.600 | 3.340 |
| FGSM | **6.820** | 0.680 | 6.300 | 0.590 | 0.740 | 0.550 | 0.480 | 0.460 | 2.380 |
| PGD[10] | **5.990** | 0.530 | 5.800 | 0.430 | 0.690 | 0.300 | 0.280 | 0.450 | 2.170 |

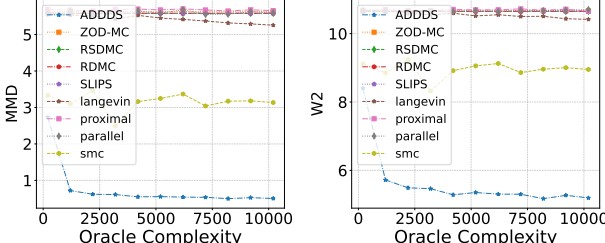

*Figure 8.* Sampling accuracies of different methods w.r.t. oracle complexity for a discontinuous potential under PGD[10] attacks. A lower score indicates a better performance.

SLIPS achieves $9.89\%$. ADDDS successfully maintains stable performance in a complex and high-dimensional posterior sampling task where some baselines even collapse completely under attack (e.g., dropping to $< 1\%$). Hence, from the perspective of parameter posterior sampling, such experimental results could provide practical evidence of robustness for ADDDS. Besides, they also provide a representative proof-of-concept demonstration.

## 5. Conclusion

Denoising diffusion sampling (DDS) is a popular approach to generate new samples with the same distribution as some

given samples. However, this approach is vulnerable to adversarial attacks, especially in the transmission stage of a decentralized setting. To address this challenge, we propose a complete set of adversarial attack and defense methodology. In the attack side, we find that even a simple Gaussian perturbation can sabotage the generation of new correct samples, let alone other advanced attack strategies. In the defense side, we develop a joint minimization model of the potential function and the local variation (LV), which can tolerate the injected adversarial perturbations. Moreover, we develop a conjugate gradient algorithm to solve the defense model. It integrates with a zeroth-order diffusion Monte Carlo method and saves computational cost.

Extensive experiments show that the proposed attack method successfully sabotages the generation of new samples against different DDS methods. On the other side, the proposed defense method successfully recovers the generated samples from the ground-truth distribution against the adversarial perturbations. In summary, theses findings not only expose a vulnerable stage in DDS, but also characterize such an adversarial perturbation by the LV. Future works may lie in exploring more vulnerable parts in different diffusion models and extend the proposed LV methodology to tackle more types of adversarial perturbations.

## Acknowledgments

This work was supported in part by the National Natural Science Foundation of China (Nos. 62332007, U22B2028, 62541606), Shenzhen Major Science and Technology Special Project (No. KJZD20240903101108011), National Joint Engineering Research Center of Network Security Detection and Protection Technology, Guangdong Key Laboratory of Data Security and Privacy Preserving, Guangdong Hong Kong Joint Laboratory for Data Security and Privacy Protection, Engineering Research Center of Trustworthy AI, Ministry of Education, and the National Cyber Security-National Science and Technology Major Project (No. 2026ZD1500303).

Code is available at https://github.com/laizhr/ADDDS.

## Impact Statement

This paper presents work whose goal is to advance the field of Machine Learning. There are many potential societal consequences of our work, none of which we feel must be specifically highlighted here.

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

# A. Theoretical Details

## A.1. Proof of Theorem 3.1

*Proof.* **Part (1):**

Let $x^*$ be a solution to (5). According to Fermat's rule, we have $\nabla V(x^*) = 0$. Then for any $x \in \mathbb{R}^d$,

$$V(x^*) + \lambda|\nabla V(x^*)| = V(x^*) \leqslant V(x) \leqslant V(x) + \lambda|\nabla V(x)|. \tag{30}$$

Hence $x^*$ is also a solution to (14).

**Part (2):**

Conversely, let $x^*$ be a solution to (14). Then for any $x \in \mathbb{R}^d$, we have

$$V(x^*) + \lambda|\nabla V(x^*)| \leqslant V(x) + \lambda|\nabla V(x)|. \tag{31}$$

Suppose $x^*$ is not a solution to (5), then there exists a solution $x^\circ \neq x^*$ to (5) such that $V(x^\circ) < V(x^*)$. According to Fermat's rule, we have $\nabla V(x^\circ) = 0$. Inserting $x = x^\circ$ into (31) yields

$$V(x^*) \leqslant V(x^*) + \lambda|\nabla V(x^*)| \leqslant V(x^\circ) + \lambda|\nabla V(x^\circ)| = V(x^\circ), \tag{32}$$

which contradicts the above-mentioned deduction $V(x^\circ) < V(x^*)$. Hence $x^*$ should also be a solution to (5).

$\square$

## A.2. Convergence of Algorithm (19)

First, we need to verify that the level set $\mathscr{L} := \{x : W(x) \leqslant W(x^{(0)})\}$ is bounded. In the literature, many analytic potential functions $V$ satisfy this bounded level set condition, like the exponential family and the quadratic family in forms of $V_m$ and $V_q$ in (42), respectively. Moreover, the LV $|\nabla V(x)|$ also satisfies the bounded level set condition if $V$ belongs to the analytic potential function family like $V_m$ and $V_q$. Since $W(x) = V(x) + \lambda|\nabla V(x)|$ with $\lambda \geqslant 0$, $W(x)$ also satisfies the bounded level set condition.

Second, we need to verify that in a neighborhood $\mathscr{N}$ of $\mathscr{L}$, $W$ has Lipschitz smoothness: there exists a constant $L_w > 0$, such that

$$\|\nabla W(x) - \nabla W(x')\|_2 \leqslant L_w \|x - x'\|_2, \quad \forall x, x' \in \mathscr{N}. \tag{33}$$

Since $\mathscr{L}$ is bounded, we can also let $\mathscr{N}$ be a closed and bounded neighborhood. If $V$ is an analytic potential function like $V_m$ and $V_q$, then its second derivatives are differentiable and bounded in the closed and bounded set $\mathscr{N}$. Hence $\partial|\nabla V(x)|$ defined in (17) is differentiable and bounded in $\mathscr{N}$ except for a zero measure set $\{x : \nabla V(x) = 0\}$. Since $\nabla W(x) = \nabla V(x) + \lambda\partial|\nabla V(x)|$, $\nabla W(x)$ is also differentiable and bounded in $\mathscr{N}$ almost everywhere (a.e.). Hence $\nabla W(x)$ is Lipschitz continuous in $\mathscr{N}$ a.e. with some Lipschitz constant $L_w$. In practical numerical computations, the iterate $x^{(l)}$ hardly hits a zero measure set like $\{x : \nabla V(x) = 0\}$ (or we could add a small distraction to the iterate to escape from this zero set), hence we could consider $W$ Lipschitz-smooth in $\mathscr{N}$.

Third, since the Wolfe line search (20) satisfies the Zoutendijk condition (Zoutendijk, 1970; Wolfe, 1969; 1971):

$$\sum_{l=1}^{\infty} \frac{((\nabla W(x^{(l)}))^\top \mathfrak{d}^{(l)})^2}{\|\mathfrak{d}^{(l)}\|_2^2} < \infty, \tag{34}$$

it follows from Theorem 4.3 of (Gilbert & Nocedal, 1992) that algorithm (19) converges in the following sense:

$$\liminf_{l \to \infty} \|\nabla W(x^{(l)})\|_2 = 0. \tag{35}$$

In practice, we strictly follow standard optimization literature (Nocedal & Wright, 2006) by setting $c_1 = 10^{-4}$ and $c_2 = 0.9$ to ensure consistent convergence.

### A.3. Proof of Theorem 3.3

**Part (a):**

*Proof.* Similar to (33), $L_{x,\epsilon}$-smoothness ($L_{x,\epsilon} > 0$) can be defined by

$$\|\nabla V(y) - \nabla V(y')\|_2 \leqslant L_{x,\epsilon}\|y - y'\|_2, \quad \forall y, y' \in \mathcal{B}_2[x;\epsilon]. \tag{36}$$

If $V(y)$ is $L_{x,\epsilon}$-smooth in the neighborhood $\mathcal{B}_2[x;\epsilon]$, it follows from Proposition A.24 of (Bertsekas, 1999) that

$$V(x+\delta) \leqslant V(x) + \nabla V(x)^\top \delta + \frac{L_{x,\epsilon}}{2}\|\delta\|_2^2, \quad \forall \|\delta\|_2 \leqslant \epsilon. \tag{37}$$

The right side of (37) is a quadratic form of $\delta$ that takes the maximum value at $\delta = \epsilon\frac{\nabla V(x)}{\|\nabla V(x)\|_2}$. Hence,

$$\max_{\|\delta\|_2 \leqslant \epsilon} V(x+\delta) \leqslant V(x) + \epsilon\|\nabla V(x)\|_2 + \frac{L_{x,\epsilon}}{2}\|\epsilon\|_2^2$$

$$\leqslant V(x) + \lambda\|\nabla V(x)\|_2 + \frac{L_{x,\epsilon}}{2}\|\epsilon\|_2^2$$

$$= W(x) + \frac{L_{x,\epsilon}}{2}\|\epsilon\|_2^2. \tag{38}$$

Moreover, according to the Equivalent Norm Theorem, the $\ell_2$ norm is equivalent to the general $\ell_p$ norm (including the $\ell_\infty$ norm). Hence for an attack budget $\|\delta\|_{\mathfrak{p}} \leqslant \epsilon$, we have $\|\delta\|_2 \leqslant \mathcal{C}_{\mathfrak{p}}\epsilon$ with some constant $\mathcal{C}_{\mathfrak{p}} > 0$. Substituting $\|\delta\|_2 \leqslant \mathcal{C}_{\mathfrak{p}}\epsilon$ into (38) leads to:

$$\max_{\|\delta\|_{\mathfrak{p}} \leqslant \epsilon} V(x+\delta) \leqslant \max_{\|\delta\|_2 \leqslant \mathcal{C}_{\mathfrak{p}}\epsilon} V(x+\delta) \leqslant W(x) + \frac{L_{x,\mathcal{C}_{\mathfrak{p}}\epsilon}}{2}\|\epsilon\|_2^2. \tag{39}$$

$\square$

**Part (b):**

If $V(y)$ is twice differentiable in $\mathcal{B}_2[x;\epsilon]$ or $\mathcal{B}_2[x;\mathcal{C}_{\mathfrak{p}}\epsilon]$, we can let

$$L_{x,\epsilon} = \max_{y \in \mathcal{B}_2[x;\epsilon]}(\varsigma_{max}(\nabla^2 V(y))) \quad \text{or} \quad L_{x,\mathcal{C}_{\mathfrak{p}}\epsilon} = \max_{y \in \mathcal{B}_2[x;\mathcal{C}_{\mathfrak{p}}\epsilon]}(\varsigma_{max}(\nabla^2 V(y))), \tag{40}$$

where $\varsigma_{max}$ denotes the largest eigenvalue of the Hessian $\nabla^2 V(y)$. Then $L_{x,\epsilon}$ or $L_{x,\mathcal{C}_{\mathfrak{p}}\epsilon}$ satisfies (38) or (39), respectively.

**Part (c):**

We present how to deduce (23) from (21) by inducing a contradiction. Let $\mathfrak{g}(x)$ and $\mathfrak{f}(x)$ be the left-hand side (after maximizing $V(x+\delta)$ over $\delta$ this becomes a function of $x$) and the middle expression of (21), respectively. Then (21) indicates that $\mathfrak{g}(x) \leqslant \mathfrak{f}(x)$ for all $x$. Suppose $x_1 \in \arg\min \mathfrak{g}(x)$ and $x_2 \in \arg\min \mathfrak{f}(x)$, and that $x_1$ is not necessarily the same as $x_2$. Then $\mathfrak{g}(x_1) \leqslant \mathfrak{f}(x_2)$ truly holds. If not, which means that $\mathfrak{g}(x_1) > \mathfrak{f}(x_2)$, then $\mathfrak{g}(x_1) > \mathfrak{f}(x_2) \geqslant \mathfrak{g}(x_2)$ (because $\mathfrak{g}(x) \leqslant \mathfrak{f}(x)$ for all $x$). This leads to $\mathfrak{g}(x_1) > \mathfrak{g}(x_2)$, which contradicts the fact that $x_1 \in \arg\min \mathfrak{g}(x)$. To summarize, $\mathfrak{g}(x_1) \leqslant \mathfrak{f}(x_2)$ truly holds even if $x_1 \neq x_2$, which verifies (23).

## B. Experimental Details and Additional Experiments

The experimental settings of this paper basically follow those of (He et al., 2024) to make fair comparisons, which are detailed in this section. The STD of the Gaussian attack is set to $\sigma = 0.01$ or $0.1$, while the perturbation budget of FGSM and PGD$^{10}$ in (7) and (8) is set to $\epsilon = 0.01$.

### B.1. 2D Gaussian Mixture Model on Different Oracle Complexities and Different Dimensionalities

A four-mode 2D GMM with the following default parameters is used in this paper:

$$
\begin{aligned}
&w: 0.1, 0.2, 0.3, 0.4, \\
&\mu: \begin{bmatrix} 0 \\ 0 \end{bmatrix}, \begin{bmatrix} 0 \\ 11 \end{bmatrix}, \begin{bmatrix} 9 \\ 9 \end{bmatrix}, \begin{bmatrix} 11 \\ 0 \end{bmatrix}, \\
&\Sigma: \begin{bmatrix} 1 & 0.5 \\ 0.5 & 1 \end{bmatrix}, \begin{bmatrix} 0.3 & -0.2 \\ -0.2 & 0.3 \end{bmatrix}, \begin{bmatrix} 1 & 0.3 \\ 0.3 & 1 \end{bmatrix}, \begin{bmatrix} 1.2 & -1 \\ -1 & 1.2 \end{bmatrix},
\end{aligned}
\tag{41}
$$

where $w$, $\mu$, and $\Sigma$ denote the weights, means, and covariances for the four modes in the 2D GMM, respectively. The hyperparameters for the compared methods are set as Table A1. The proposed ADDDS uses the same hyperparameters as those of ZOD-MC, since the former integrates the latter in the recovering stage. The regularization strength for ADDDS is set to a trivial value $\lambda = 1$. Additional experimental results on different oracle complexities and different attack schemes are provided in Figures A2, A4, A5, A6, A7. Figure A1 shows that the runtime of ADDDS ranks amongst other baselines and keeps steady as the oracle complexity increases.

As for dimensionality dependence, given a fixed dimensionality $d$, we first randomly sample $u \in U[0,1]^d$ and $\varsigma^2 \in U[0.3, 1.3]$, then let $\mu = 12 \cdot \frac{u}{\|u\|_2}$. This yields a Gaussian distribution $\mathcal{N}(\mu, \varsigma^2 \mathbb{I}_d)$. This procedure is repeated for five times to form an equally-weighted five-mode GMM. Then the experiments on different dimensionalities and different attack schemes are conducted and shown in Figures 4(b) and A3.

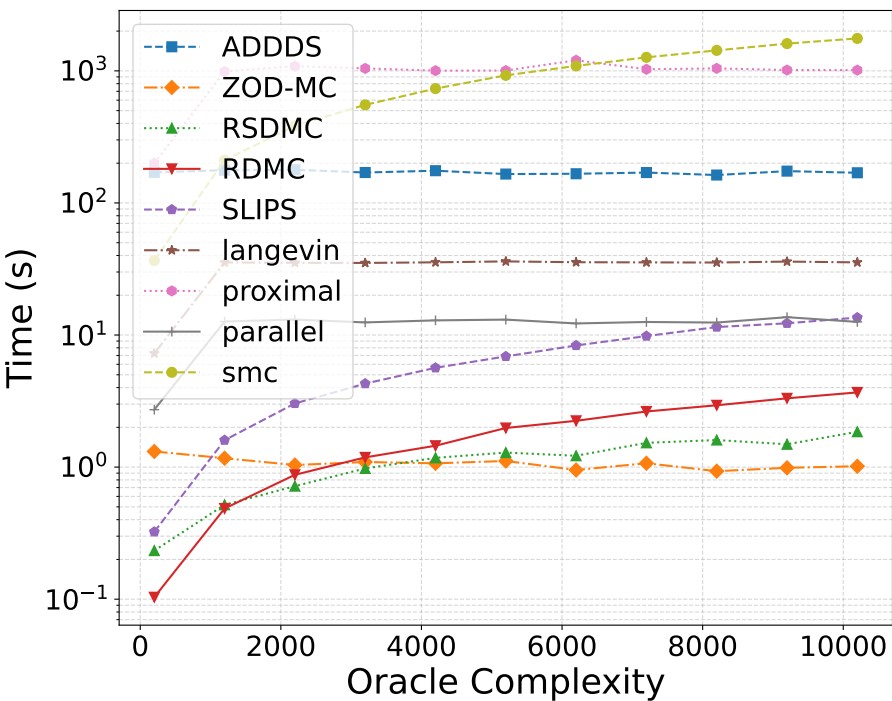

*Figure A1.* Runtimes of different methods w.r.t. oracle complexity for the 2D GMM experiment.

*Table A1.* Hyperparameters of different methods for the oracle complexity and dimensionality dependence experiments. These hyperparameters include the terminal time of the diffusion process $T$, the number of timestamps $N$, the early stopping parameter $\zeta$, the inner-loop step size of Markov Chain Monte Carlo (MCMC), the number of MCMC samples, the number of MCMC steps, the number of Markov chains. $K$ and $M$ indicate the current oracle complexity and a matched oracle complexity, respectively. 2 recursions per score evaluation are used for RSDMC.

| Method | $T$ | $N$ | $\zeta$ | Step Size | N-MCMC | Num Steps | N-Chains |
|---|---|---|---|---|---|---|---|
| ADDDS | 2 | 25 | 5e-3 | - | $K$ | - | - |
| ZOD-MC | 2 | 25 | 5e-3 | - | $K$ | - | - |
| RDMC | 2 | 25 | 5e-2 | 0.01 | 1000 | $K/1000$ | - |
| RSDMC | 2 | 25 | 5e-2 | 0.01 | $K^{1/4}$ | $K^{1/4}$ | - |
| SLIPS | 1 | 25 | 6.62e-3 | Adaptive | 1000 | $K/1000$ | - |
| AIS | - | - | - | Adaptive | - | M | 512 |
| SMC | - | - | - | Adaptive | - | M | 512 |
| Langevin | - | - | - | 0.01 | - | M | - |
| Proximal | - | - | - | 1/5 | - | M | - |
| Parallel | - | - | - | 0.01 | - | M | 512 |

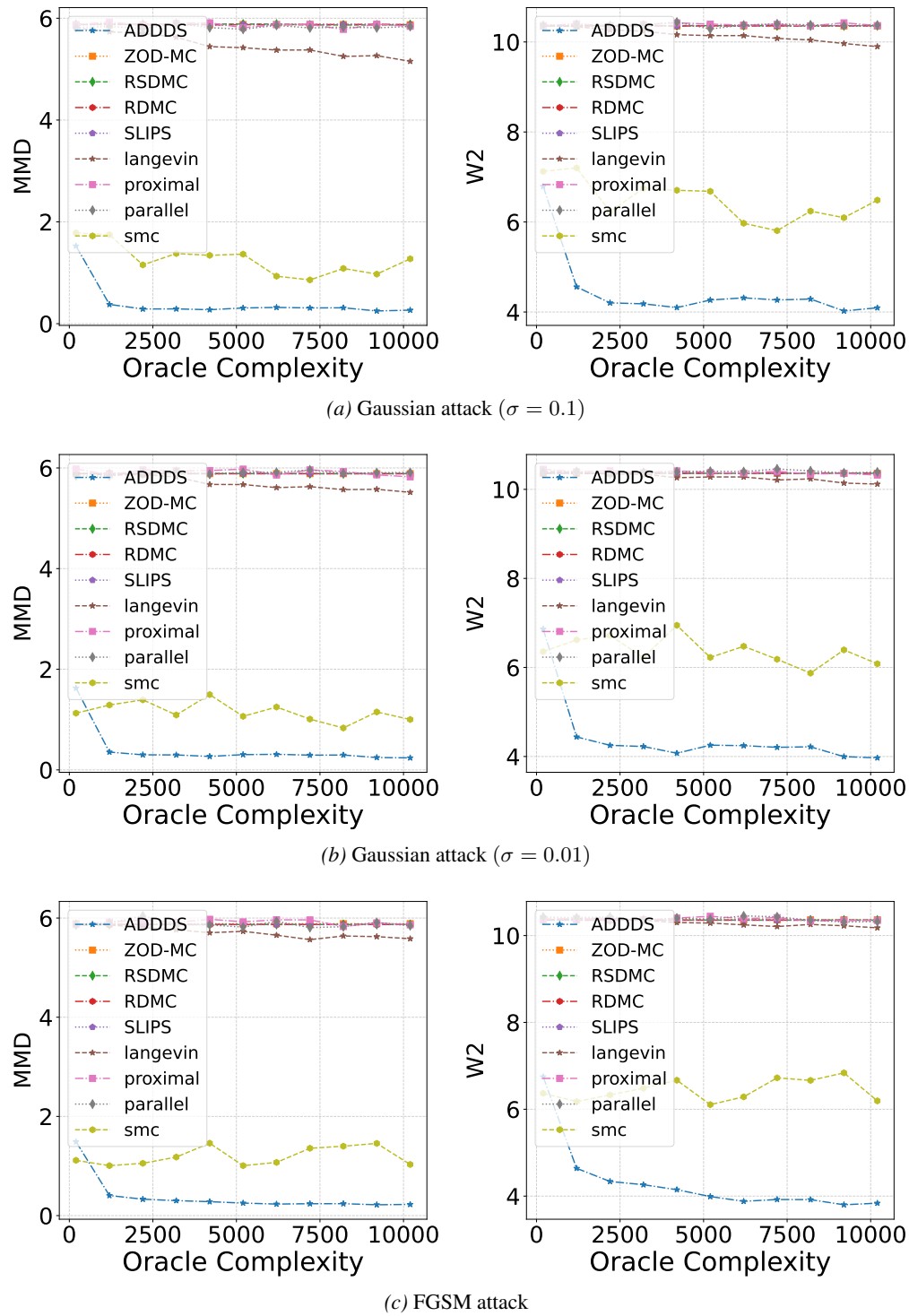

*(a)* Gaussian attack ($\sigma = 0.1$)

*(b)* Gaussian attack ($\sigma = 0.01$)

*(c)* FGSM attack

*Figure A2.* Sampling accuracies of different methods w.r.t. oracle complexity for Gaussian mixtures under different adversarial attacks. A lower score indicates a better performance.

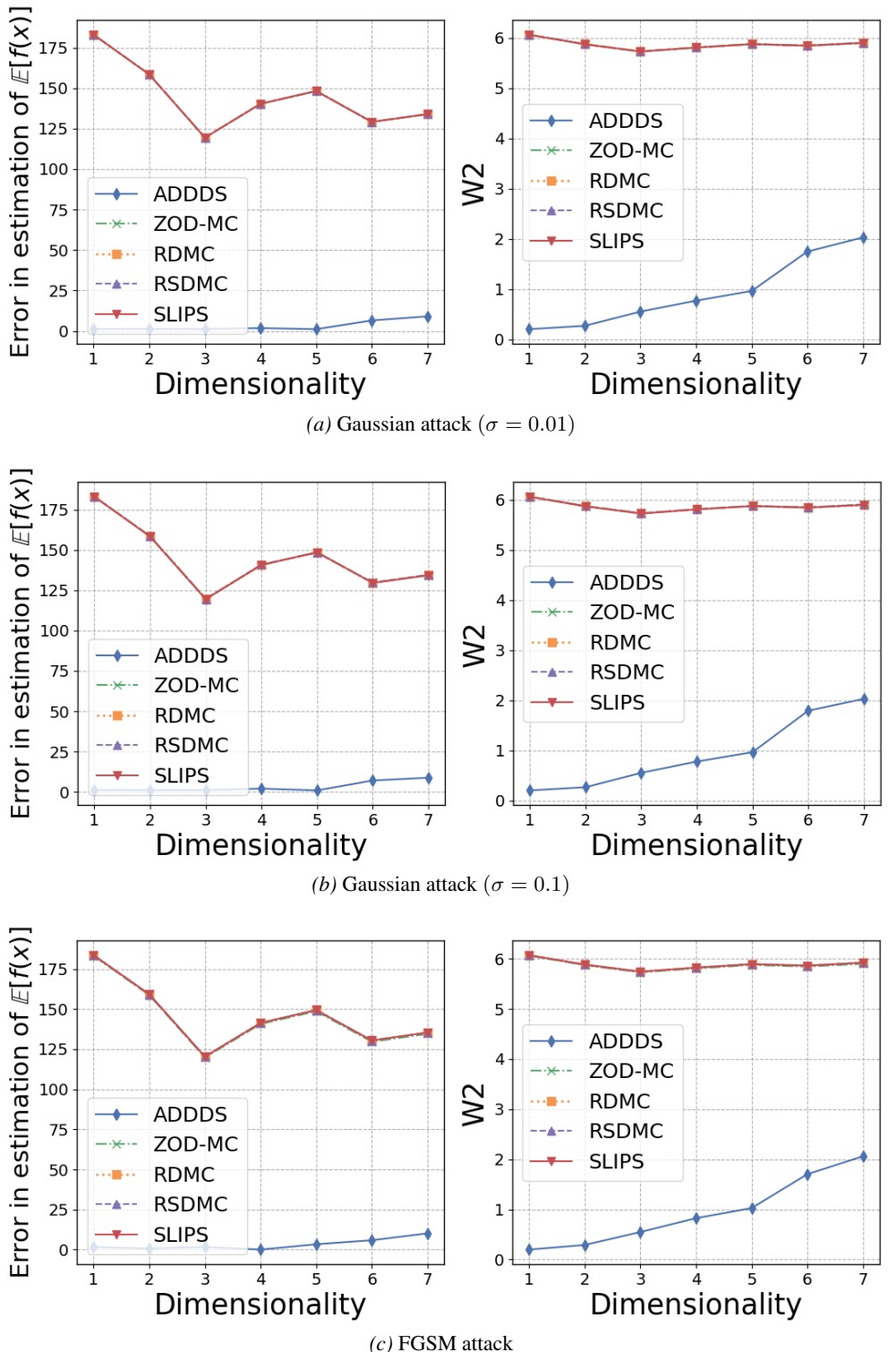

(a) Gaussian attack ($\sigma = 0.01$)

(b) Gaussian attack ($\sigma = 0.1$)

(c) FGSM attack

*Figure A3.* Sampling accuracies of different methods w.r.t. dimensionality for Gaussian mixtures under different adversarial attacks. A lower score indicates a better performance. The plots of other baselines mutually overlap.

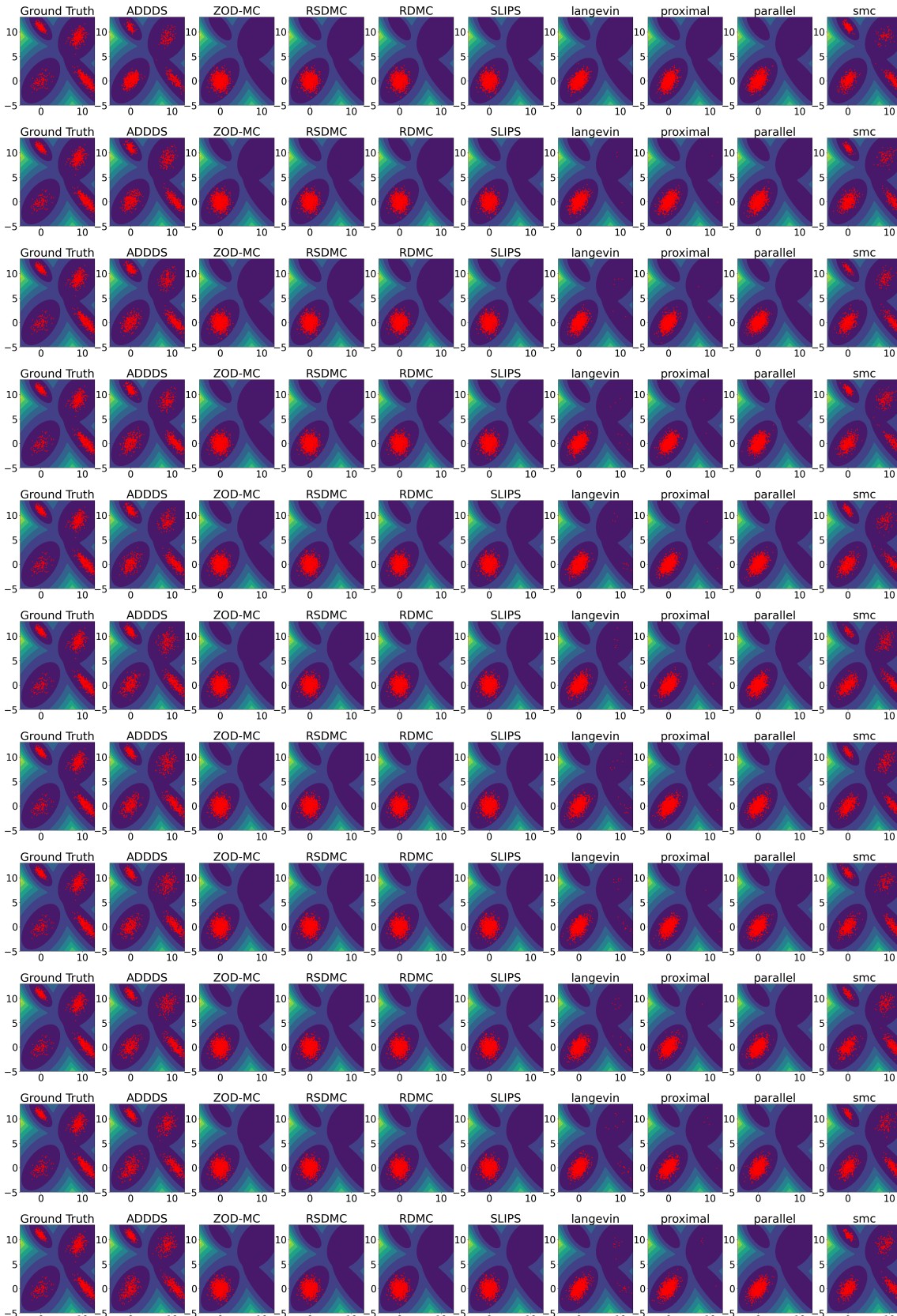

*Figure A4.* Samples generated by different methods from an asymmetric and unbalanced Gaussian mixture under Gaussian attacks ($\sigma = 0.01$). From top to bottom: oracle complexity= 200, 1200, 2200, 3200, 4200, 5200, 6200, 7200, 8200, 9200, 10200.

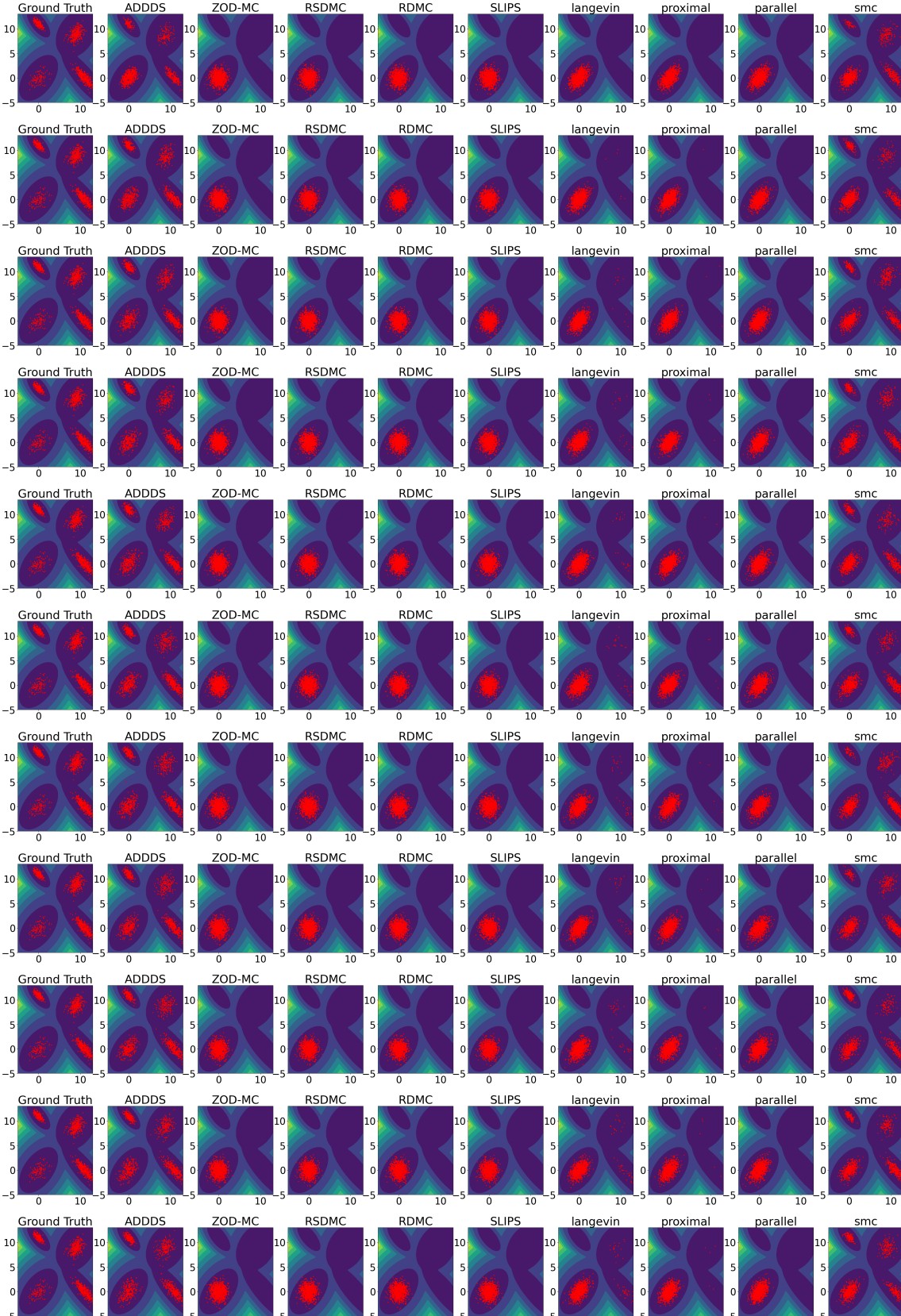

*Figure A5.* Samples generated by different methods from an asymmetric and unbalanced Gaussian mixture under Gaussian attacks ($\sigma = 0.1$). From top to bottom: oracle complexity= 200, 1200, 2200, 3200, 4200, 5200, 6200, 7200, 8200, 9200, 10200.

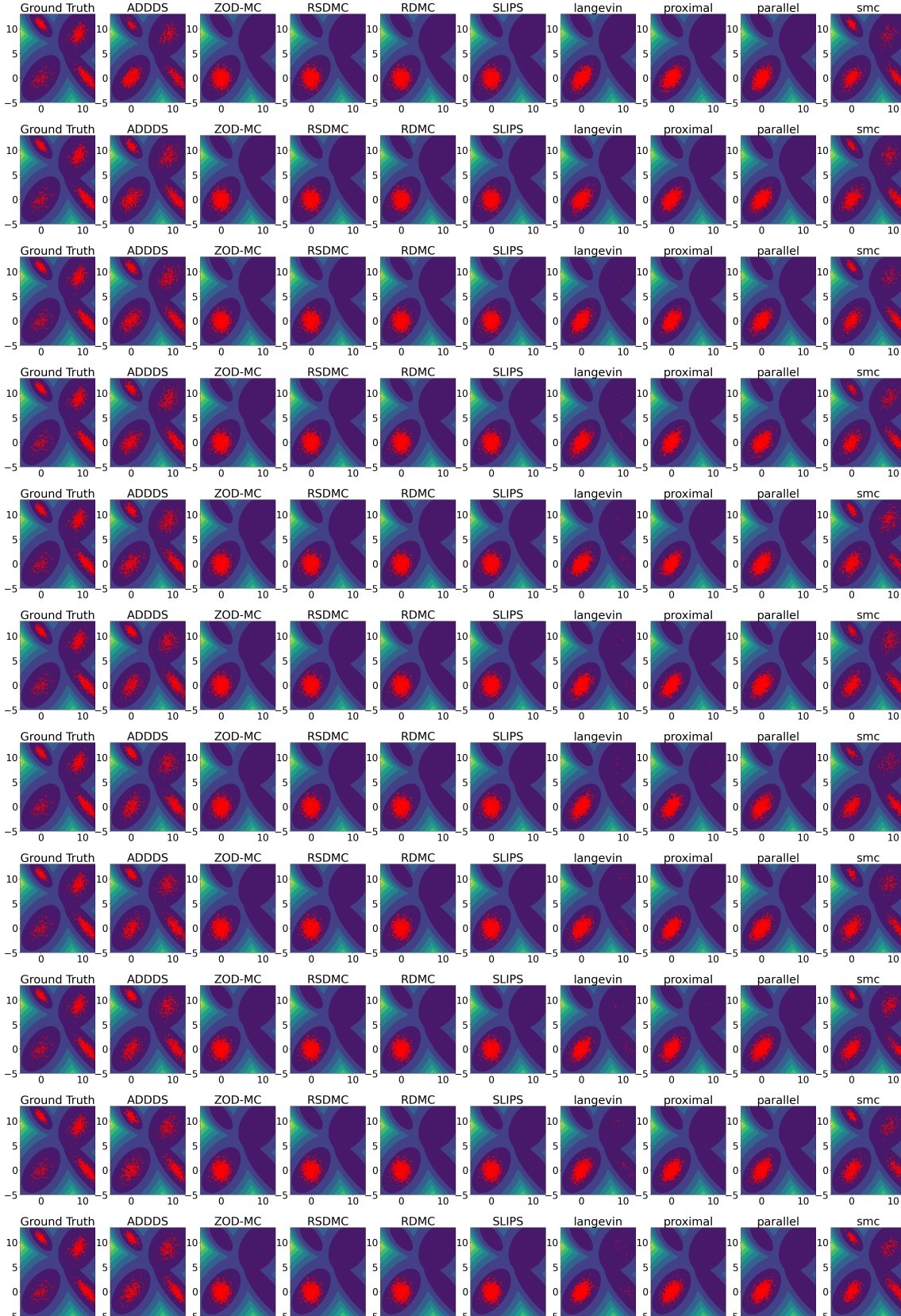

*Figure A6.* Samples generated by different methods from an asymmetric and unbalanced Gaussian mixture under FGSM attacks. From top to bottom: oracle complexity= 200, 1200, 2200, 3200, 4200, 5200, 6200, 7200, 8200, 9200, 10200.

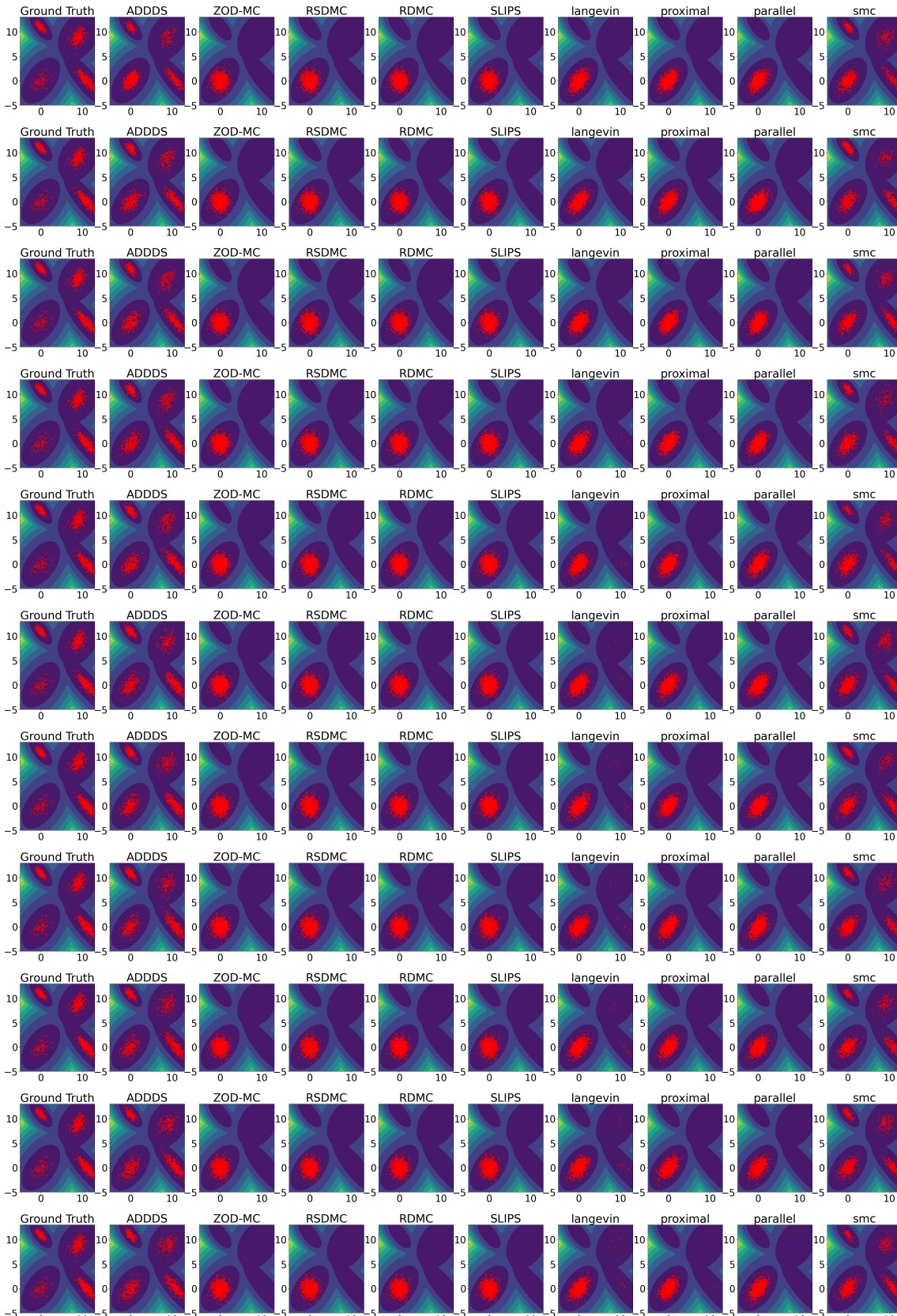

*Figure A7.* Samples generated by different methods from an asymmetric and unbalanced Gaussian mixture under $\text{PGD}^{10}$ attacks. From top to bottom: oracle complexity= 200, 1200, 2200, 3200, 4200, 5200, 6200, 7200, 8200, 9200, 10200.

## B.2. Robustness to Different Radii

The hyperparameters for the compared methods are set as Table A2, where ADDDS uses the same hyperparameters as those of ZOD-MC. The experiments on different dimensionalities and different attack schemes are conducted and shown in Figures 7, A8, A9, A10, A11, and A12.

*Table A2.* Hyperparameters of different methods for the mode separation experiment. These hyperparameters include the terminal time of the diffusion process $T$, the number of timestamps $N$, the early stopping parameter $\zeta$, the inner-loop step size of Markov Chain Monte Carlo (MCMC), the number of MCMC samples, the number of MCMC steps, the number of Markov chains. $K$ and $M$ indicate the current oracle complexity and a matched oracle complexity, respectively.

| Method | $T$ | $N$ | $\zeta$ | Step Size | N-MCMC | Num Steps | N-Chains |
|---|---|---|---|---|---|---|---|
| ADDDS | 10 | 50 | 5e-3 | - | $K$ | - | - |
| ZOD-MC | 10 | 50 | 5e-3 | - | $K$ | - | - |
| RDMC | 2 | 50 | 5e-2 | 0.01 | 1000 | $K/1000$ | - |
| RSDMC | 2 | 50 | 5e-2 | 0.01 | $K^{1/4}$ | $K^{1/4}$ | - |
| SLIPS | 1 | 50 | 6.62e-3 | Adaptive | 1000 | $K/1000$ | - |
| AIS | - | - | - | Adaptive | - | M | 512 |
| SMC | - | - | - | Adaptive | - | M | 512 |
| Langevin | - | - | - | 0.01 | - | M | - |
| Proximal | - | - | - | 1/40 | - | M | - |
| Parallel | - | - | - | 0.01 | - | M | 512 |

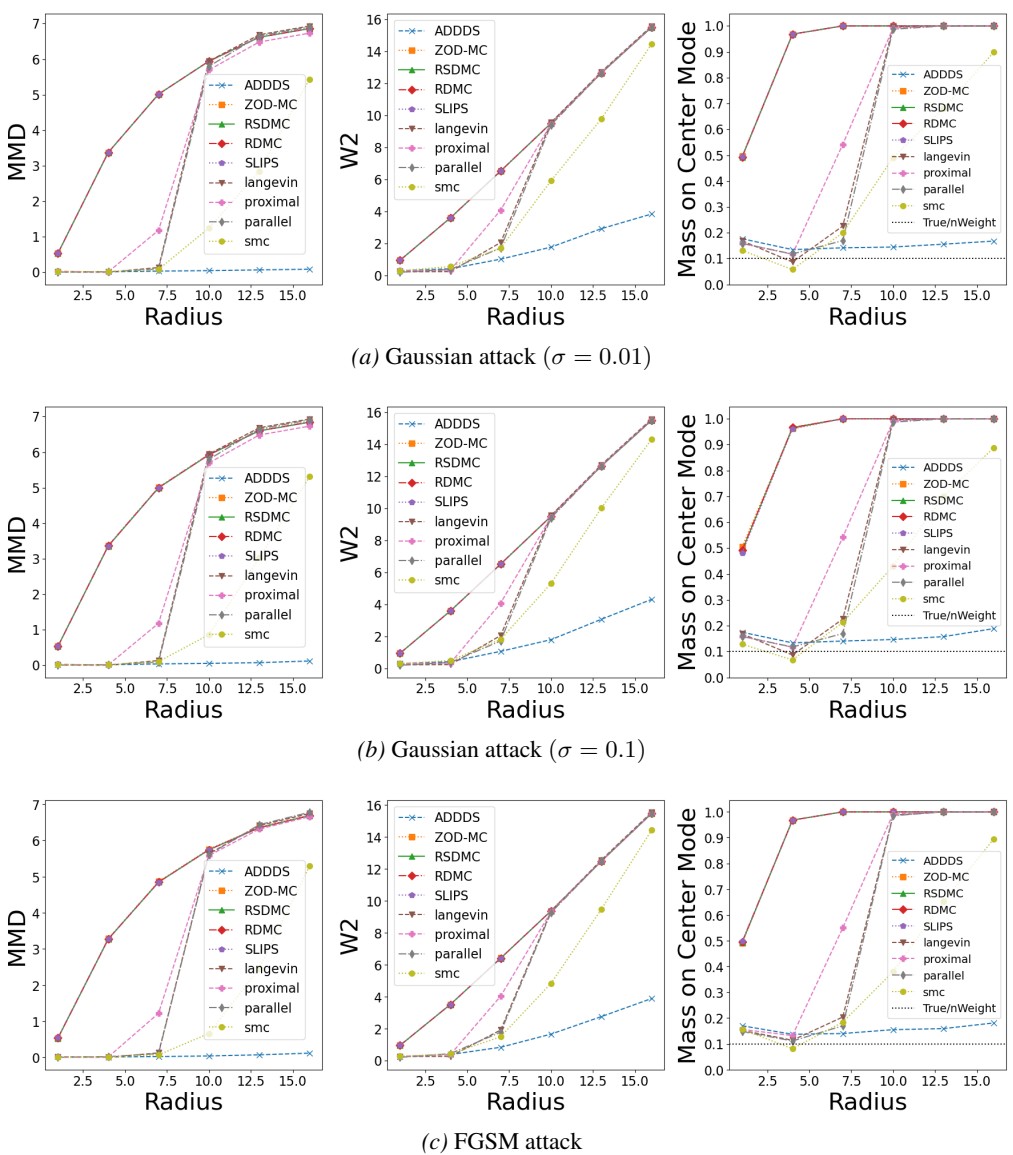

*(a)* Gaussian attack ($\sigma = 0.01$)

*(b)* Gaussian attack ($\sigma = 0.1$)

*(c)* FGSM attack

*Figure A8.* Sampling accuracies of different methods w.r.t. mode separation for Gaussian mixtures under different adversarial attacks. A lower score indicates a better performance.

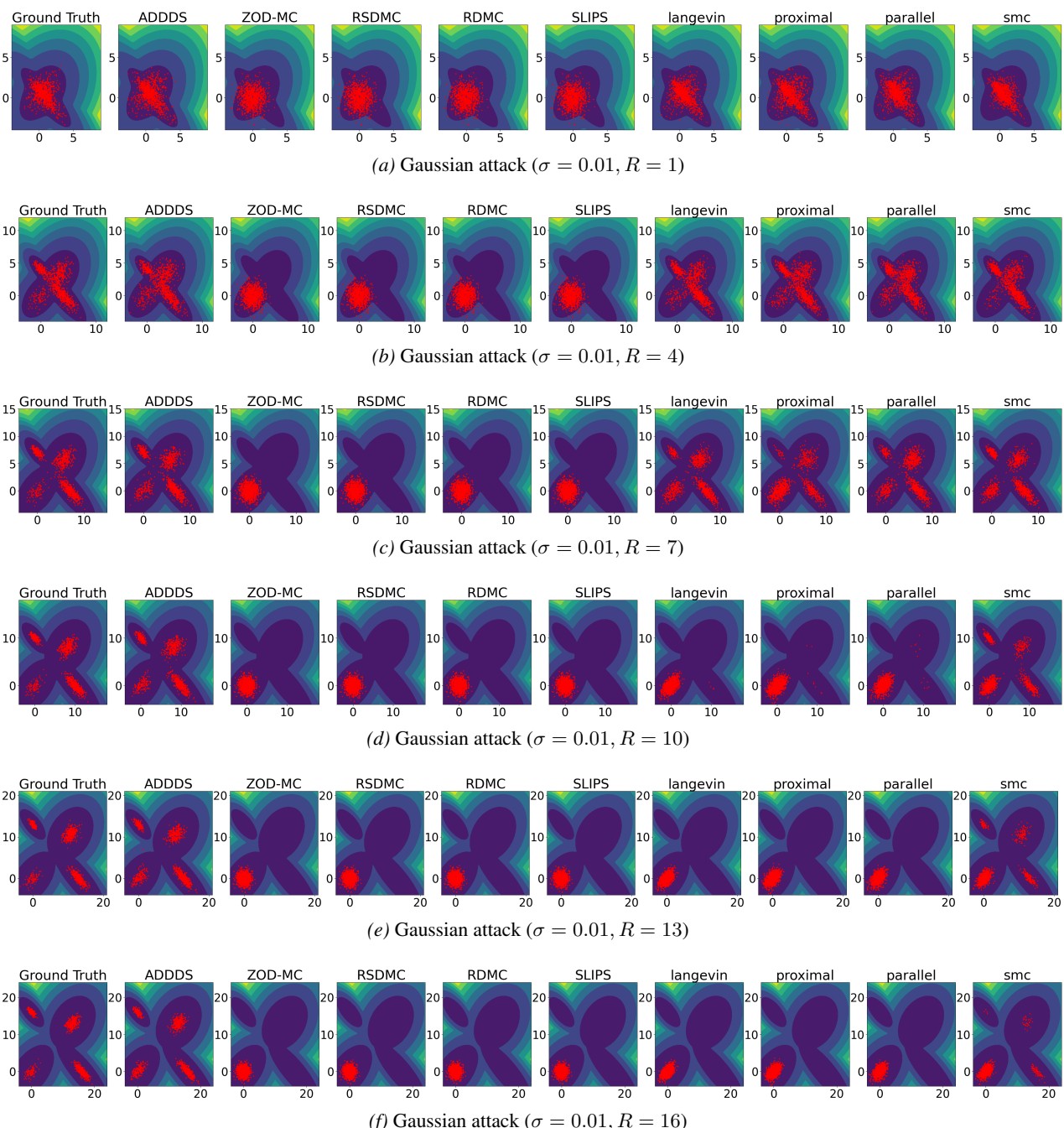

*(a)* Gaussian attack ($\sigma = 0.01, R = 1$)

*(b)* Gaussian attack ($\sigma = 0.01, R = 4$)

*(c)* Gaussian attack ($\sigma = 0.01, R = 7$)

*(d)* Gaussian attack ($\sigma = 0.01, R = 10$)

*(e)* Gaussian attack ($\sigma = 0.01, R = 13$)

*(f)* Gaussian attack ($\sigma = 0.01, R = 16$)

*Figure A9.* Samples generated by different methods for Gaussian mixtures with different radii under Gaussian attacks ($\sigma = 0.01$).

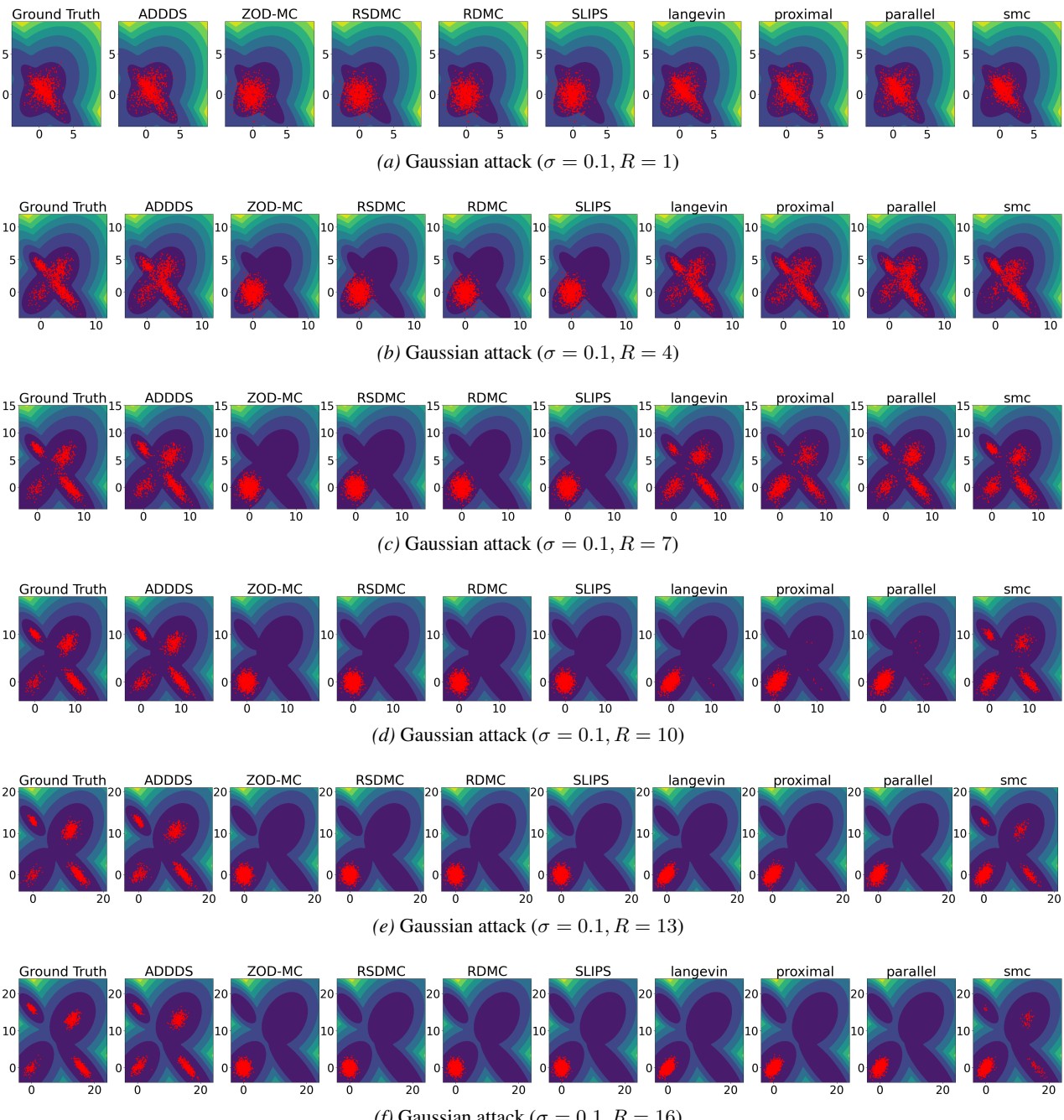

*(a)* Gaussian attack ($\sigma = 0.1, R = 1$)

*(b)* Gaussian attack ($\sigma = 0.1, R = 4$)

*(c)* Gaussian attack ($\sigma = 0.1, R = 7$)

*(d)* Gaussian attack ($\sigma = 0.1, R = 10$)

*(e)* Gaussian attack ($\sigma = 0.1, R = 13$)

*(f)* Gaussian attack ($\sigma = 0.1, R = 16$)

*Figure A10.* Samples generated by different methods for Gaussian mixtures with different radii under Gaussian attacks ($\sigma = 0.1$).

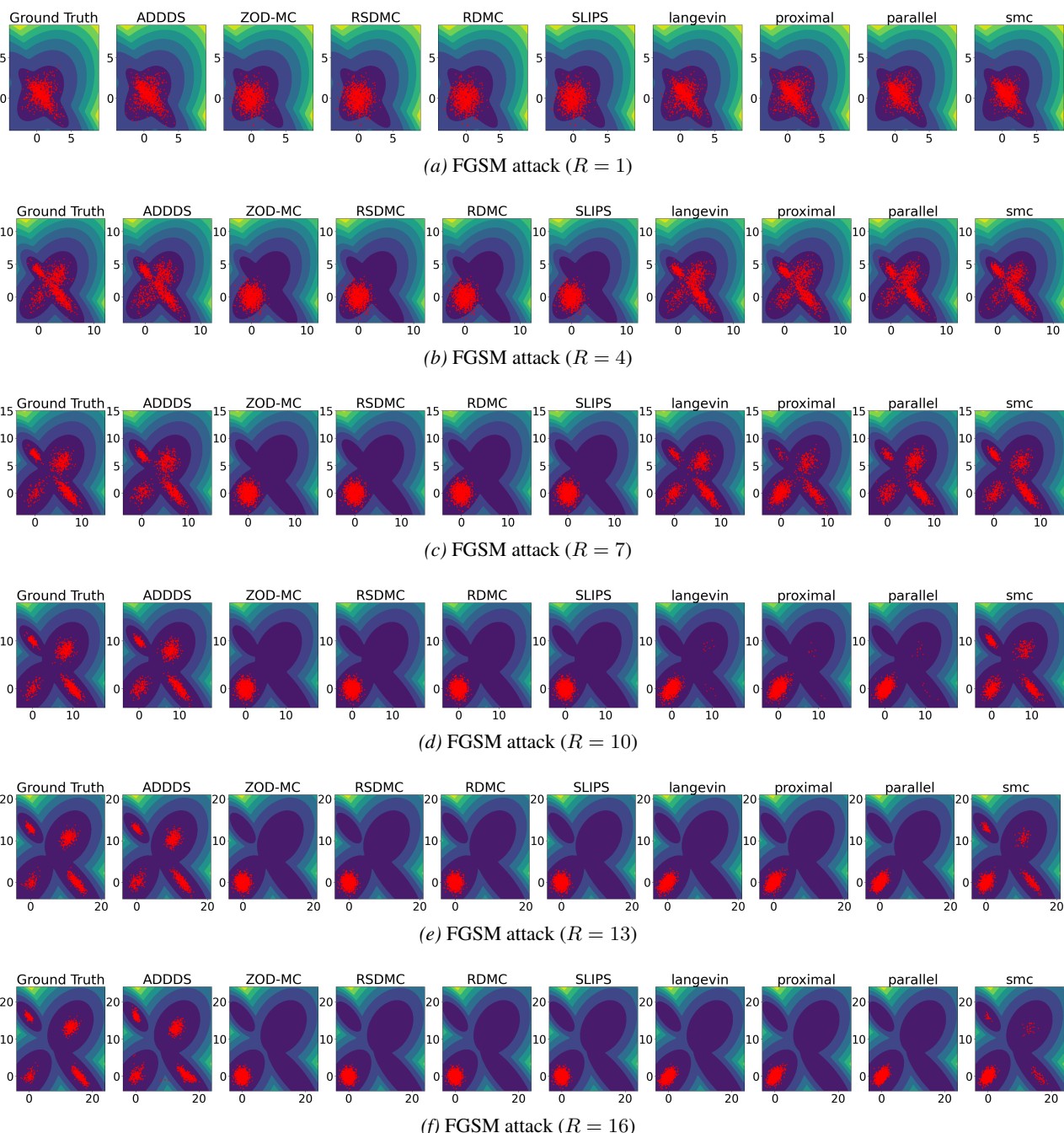

*Figure A11.* Samples generated by different methods for Gaussian mixtures with different radii under FGSM attacks.

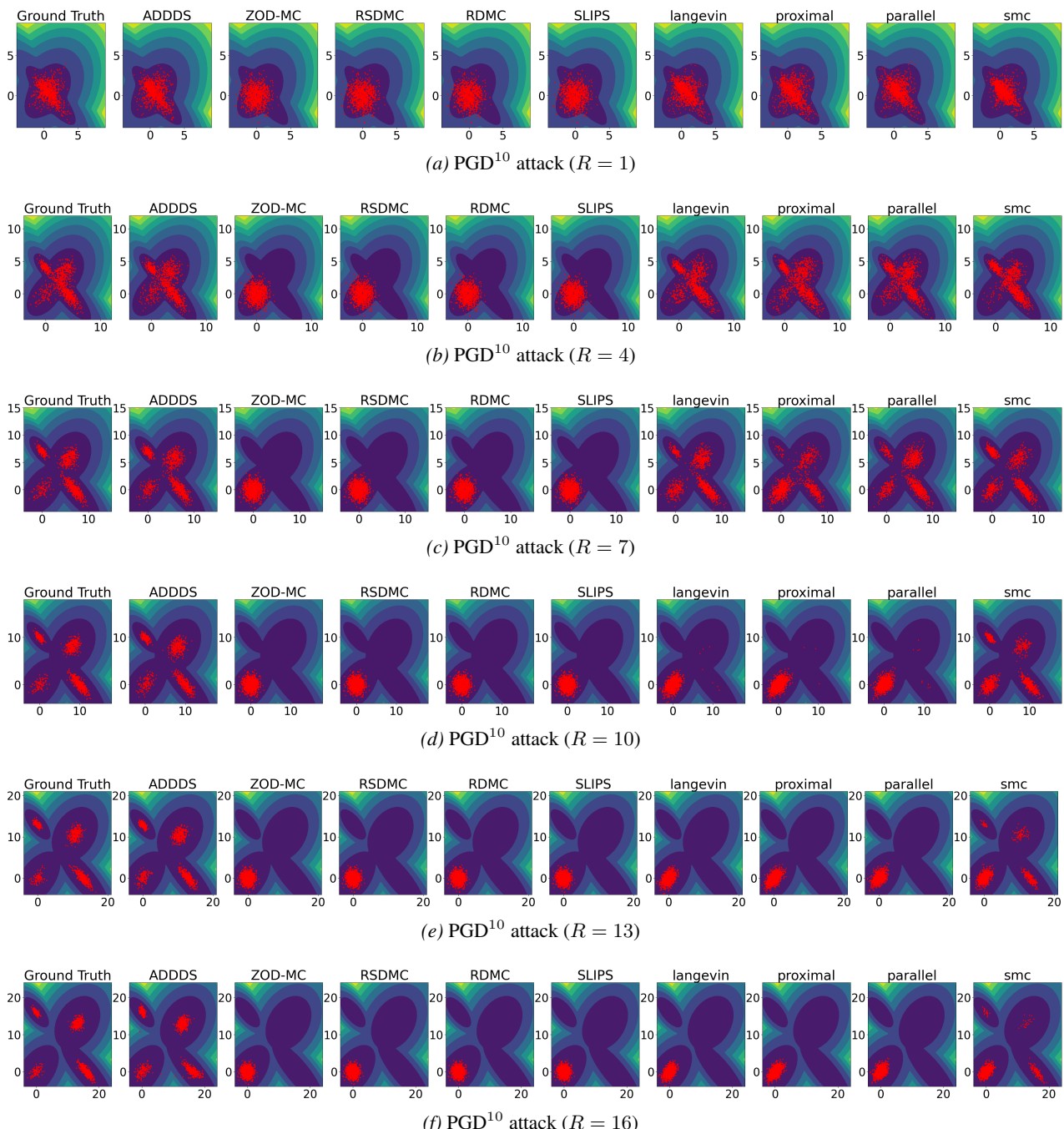

*Figure A12.* Samples generated by different methods for Gaussian mixtures with different radii under PGD$^{10}$ attacks.

## B.3. Discontinuous Potential

The hyperparameters for the compared methods are set as Table A1. The experiments for the discontinuous potential on different attack schemes are conducted and shown in Figures 8, 5, A13, A14, A15, A16, and A17.

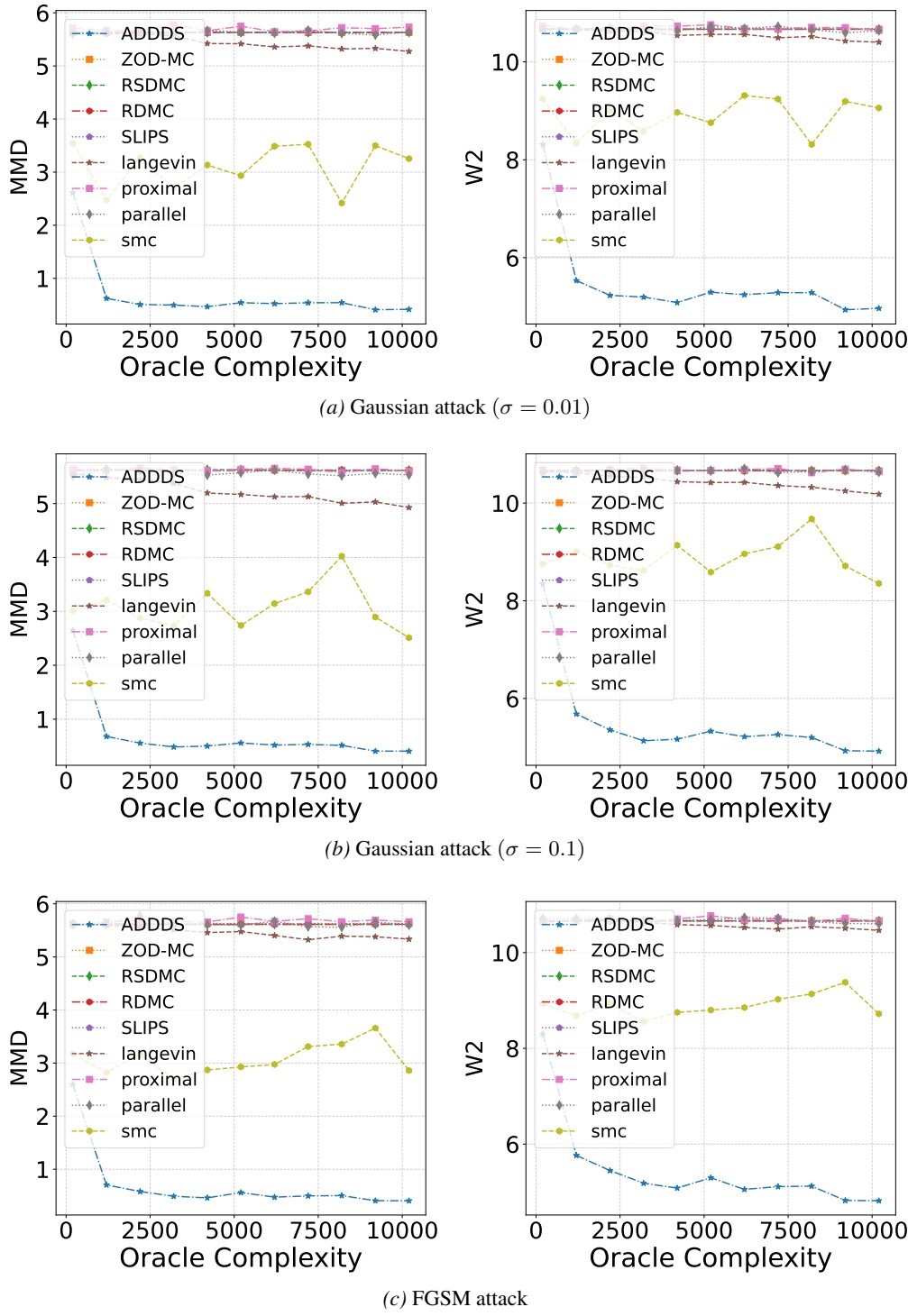

*(a)* Gaussian attack ($\sigma = 0.01$)

*(b)* Gaussian attack ($\sigma = 0.1$)

*(c)* FGSM attack

*Figure A13.* Sampling accuracies of different methods w.r.t. mode separation for a discontinuous potential under different adversarial attacks. A lower score indicates a better performance.

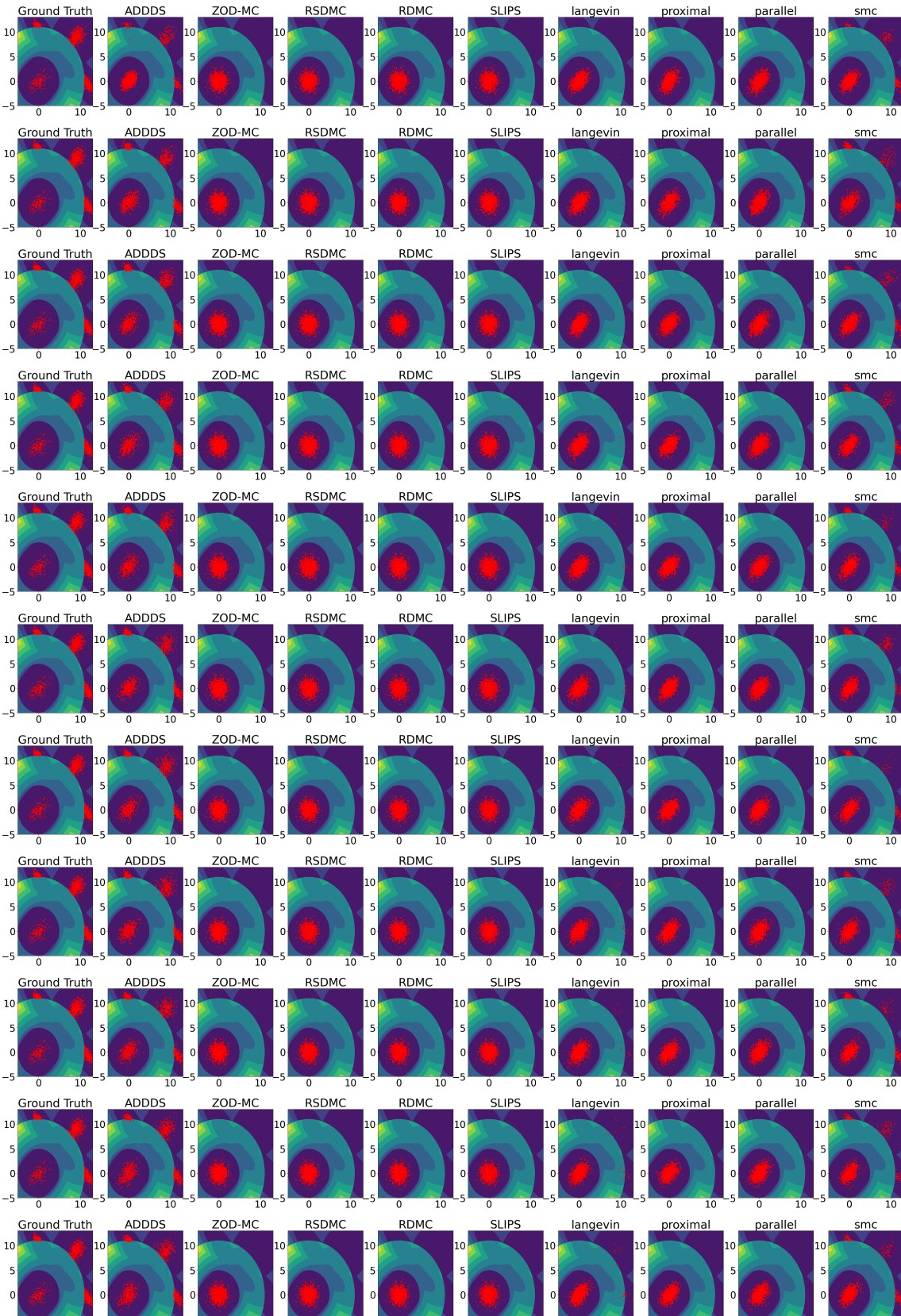

*Figure A14.* Samples generated by different methods from a discontinuous potential under Gaussian attacks ($\sigma = 0.01$). From top to bottom: oracle complexity= 200, 1200, 2200, 3200, 4200, 5200, 6200, 7200, 8200, 9200, 10200.

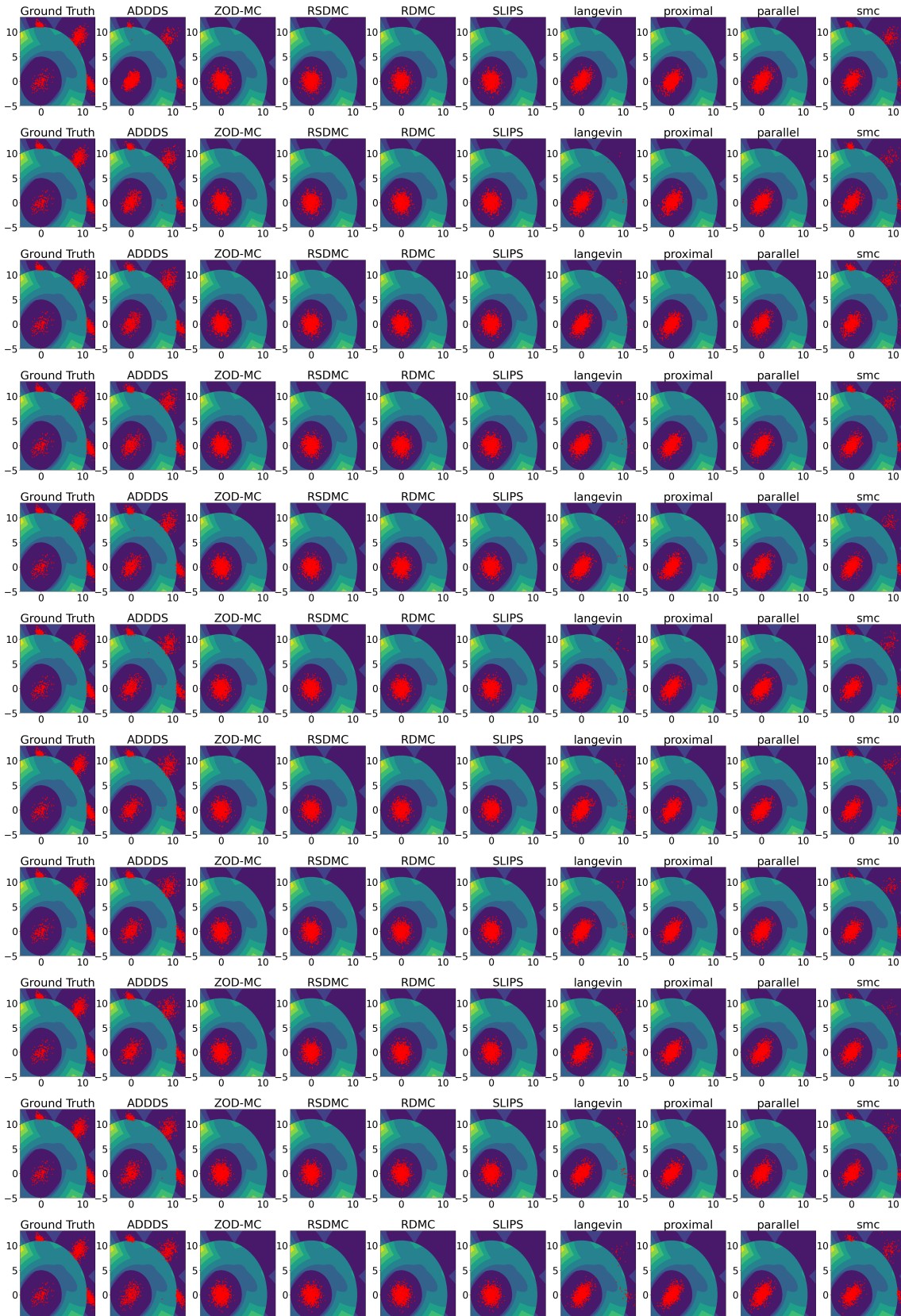

*Figure A15.* Samples generated by different methods from a discontinuous potential under Gaussian attacks ($\sigma = 0.1$). From top to bottom: oracle complexity= 200, 1200, 2200, 3200, 4200, 5200, 6200, 7200, 8200, 9200, 10200.

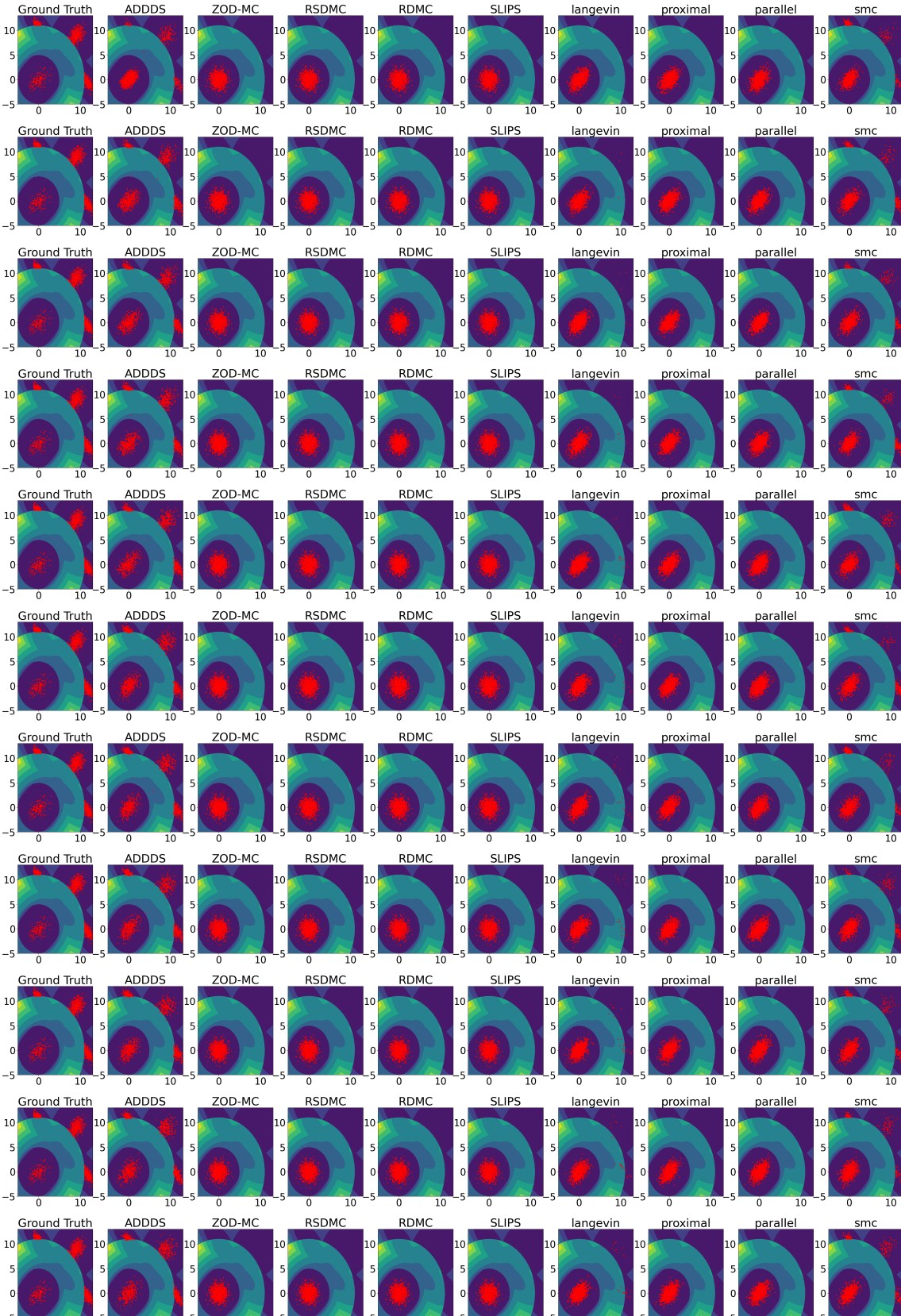

*Figure A16.* Samples generated by different methods from a discontinuous potential under FGSM attacks. From top to bottom: oracle complexity= 200, 1200, 2200, 3200, 4200, 5200, 6200, 7200, 8200, 9200, 10200.

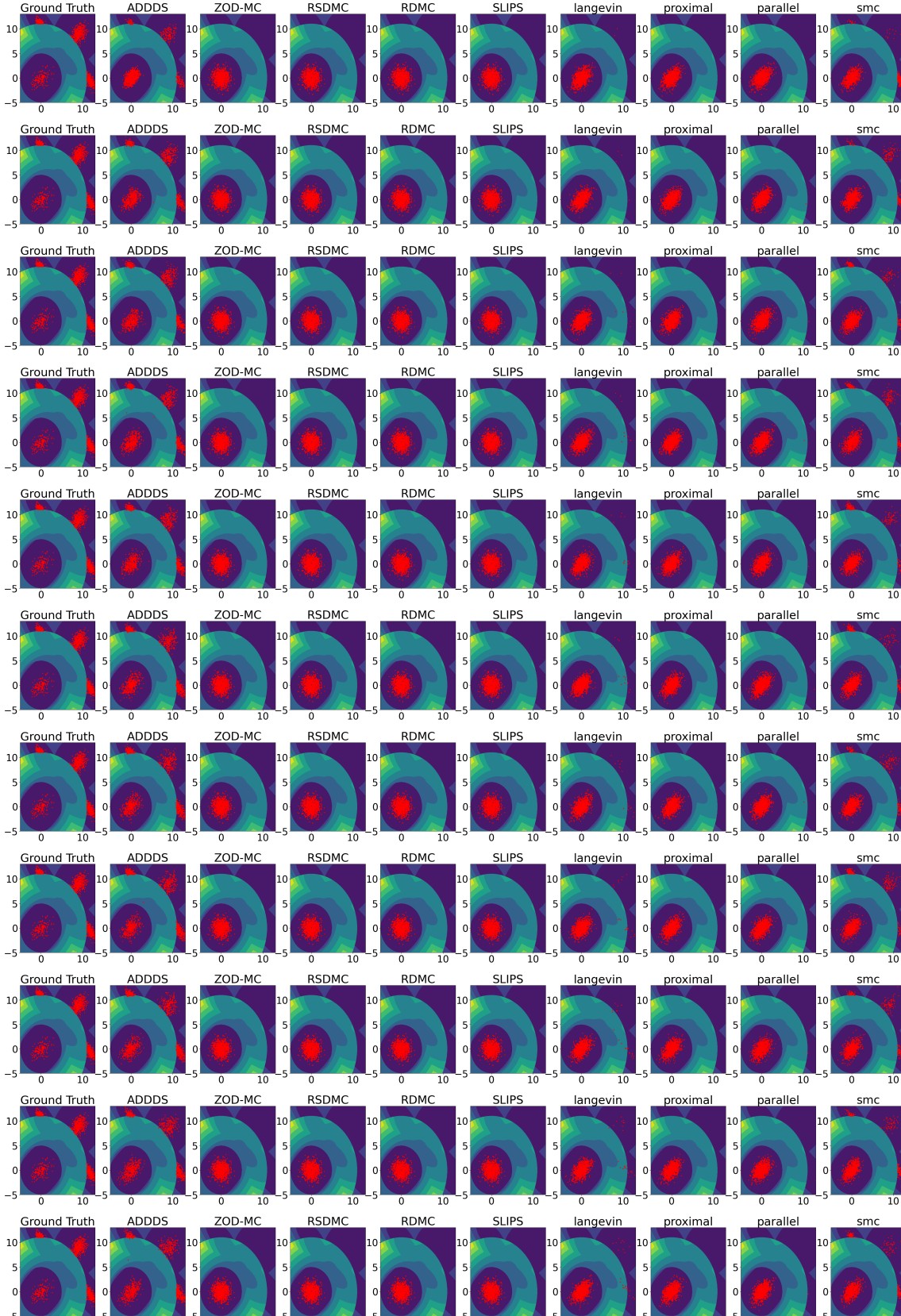

*Figure A17.* Samples generated by different methods from a discontinuous potential under PGD$^{10}$ attacks. From top to bottom: oracle complexity= 200, 1200, 2200, 3200, 4200, 5200, 6200, 7200, 8200, 9200, 10200.

## B.4. Müller-Brown Potential

The modified Müller-Brown potential used in this paper is:

$$
\begin{aligned}
V(x,y) :=& \beta(V_m(x,y) + V_q(x,y)), \\
V_m(x,y) :=& -170\exp(-6.5(x+0.5)^2 + 11(x+0.5)(y-1.5) - 6.5(y-1.5)^2) \\
& -100\exp(-x^2 - 10(y-0.5)^2) - 200\exp(-(x-1)^2 - 10y^2) \\
& +15\exp(0.7(x+1)^2 + 0.6(x+1)(y-1) + 0.7(y-1)^2), \\
V_q(x,y) :=& 35.0136(x-x_c^*)^2 + 59.8399(y-y_c^*)^2,
\end{aligned} \tag{42}
$$

where $V_m$ is the original Müller-Brown potential, and $V_q$ is an introduced correction with $(x_c^*, y_c^*)$ being the center for the middle well of $V_m$. $V_q$ makes the depths of all three wells of $V$ better leveled to each order. The scaling parameter is set to $\beta = 0.1$ by default.

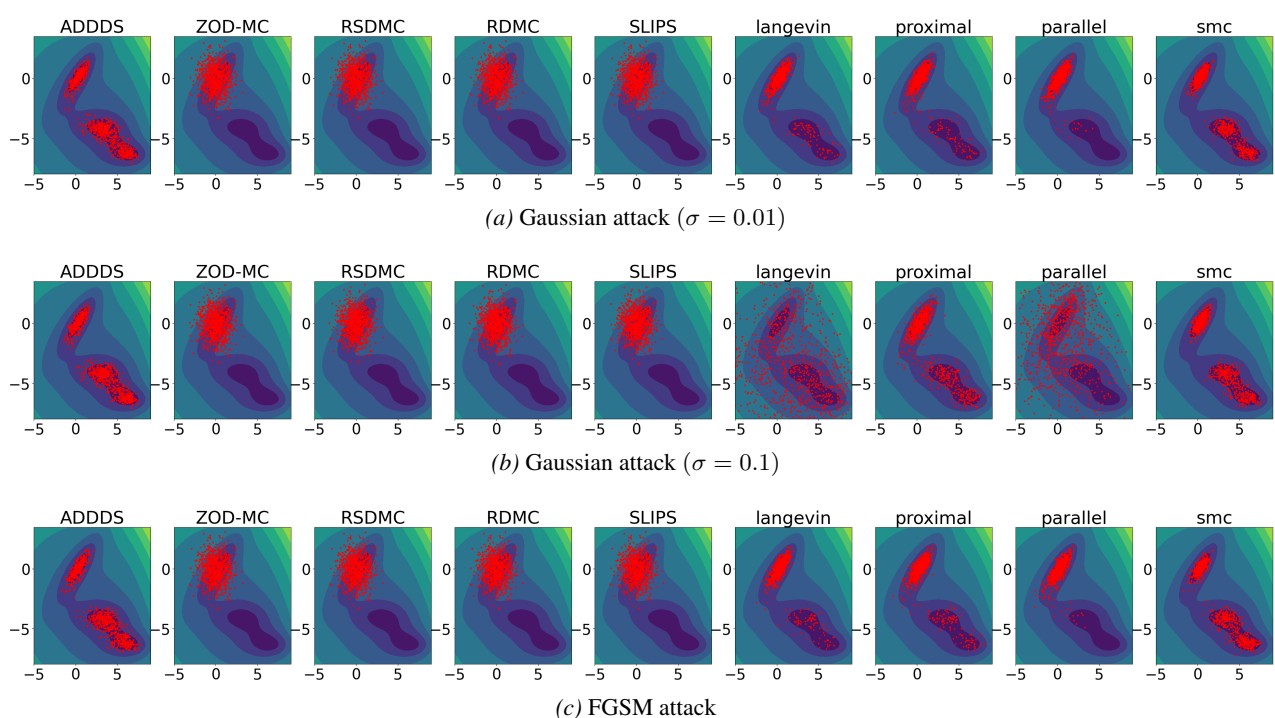

*(a)* Gaussian attack ($\sigma = 0.01$)

*(b)* Gaussian attack ($\sigma = 0.1$)

*(c)* FGSM attack

*Figure A18.* Samples generated by different methods from the Müller-Brown potential under different adversarial attacks.

## C. Additional Experiments

### C.1. Ablation Studies for Hyperparameters

We conduct ablation studies with 2D GMM under PGD[10] attack (oracle complexity= 1200) for several main hyperparameters of ADDDS: the LV regularization strength $\lambda$, the perturbation budget $\epsilon$, and the number of total sampling steps $N$. Table C1 shows that ADDDS achieves stable performance when $\lambda = 1 \sim 5$, thus $\lambda = 1$ is used for ADDDS. Table C2 shows that ADDDS is robust against different levels of $\epsilon$, while the non-defense version ZOD-MC deteriorates significantly even under the lowest perturbation $\epsilon = 0.01$. Table C3 shows that ADDDS also achieves stable performance across various settings of $N$. In summary, ADDDS is robust to both adversarial perturbations and hyperparameter change.

### C.2. Comparison with Gradient Clipping and Predictor-corrector Sampler

There are also intrinsic defense schemes for DDS, such as the gradient clipping for score estimation and the predictor-corrector sampler (Song et al., 2021). They are also taken into comparisons for the GMM model under different attacks in

*Table C1.* Ablation study for the regularization strength $\lambda$ of ADDDS under $PGD^{10}$ attack.

| Oracle | 200 | 1200 | 2200 | 3200 | 4200 | 5200 | 6200 | 7200 | 8200 | 9200 | 10200 |
|---|---|---|---|---|---|---|---|---|---|---|---|
| | | | | | | MMD | | | | | |
| $\lambda = 0.0$ | 5.876 | 5.883 | 5.883 | 5.764 | 5.931 | 5.646 | 5.892 | 5.910 | 4.880 | 4.741 | 4.478 |
| $\lambda = 0.1$ | 2.004 | 1.297 | 1.189 | 1.266 | 1.437 | 1.301 | 1.213 | 1.404 | 1.186 | 1.313 | 1.209 |
| $\lambda = 0.5$ | 1.878 | 0.643 | 0.574 | 0.561 | 0.480 | 0.473 | 0.572 | 0.523 | 0.534 | 0.496 | 0.445 |
| $\lambda = 1.0$ | **1.552** | **0.402** | **0.332** | **0.342** | **0.311** | **0.316** | **0.311** | **0.298** | **0.304** | **0.271** | **0.253** |
| $\lambda = 5.0$ | 1.431 | 0.398 | 0.351 | 0.334 | 0.325 | 0.327 | 0.336 | 0.349 | 0.310 | 0.298 | 0.326 |
| $\lambda = 10.0$ | 1.924 | 0.719 | 0.683 | 0.564 | 0.431 | 0.677 | 0.592 | 0.710 | 0.580 | 0.541 | 0.578 |
| | | | | | | W2 | | | | | |
| $\lambda = 0.0$ | 10.119 | 10.360 | 10.329 | 10.272 | 10.316 | 10.398 | 10.292 | 9.997 | 10.016 | 10.194 | 10.321 |
| $\lambda = 0.1$ | 8.566 | 7.898 | 7.210 | 7.811 | 7.519 | 7.330 | 7.012 | 7.111 | 6.947 | 6.870 | 6.950 |
| $\lambda = 0.5$ | 7.193 | 5.974 | 5.826 | 5.791 | 5.837 | 5.772 | 5.980 | 5.317 | 5.557 | 5.892 | 5.419 |
| $\lambda = 1.0$ | **6.747** | **4.631** | **4.337** | **4.279** | **4.149** | **4.287** | **3.872** | **3.981** | **3.759** | **3.810** | **3.925** |
| $\lambda = 5.0$ | 6.431 | 4.988 | 5.130 | 5.111 | 4.808 | 4.975 | 5.020 | 5.219 | 4.871 | 4.772 | 4.671 |
| $\lambda = 10.0$ | 8.909 | 6.900 | 6.218 | 6.367 | 6.211 | 6.471 | 6.410 | 6.198 | 6.509 | 6.172 | 6.084 |

*Table C2.* Ablation study for the perturbation budget $\epsilon$ under $PGD^{10}$ attack.

| | $\epsilon$ | 0.00 | 0.01 | 0.03 | 0.05 | 0.10 | 0.15 | 0.20 |
|---|---|---|---|---|---|---|---|---|
| MMD | ZOD-MC | 1.489 | 5.889 | 5.876 | 5.892 | 5.879 | 5.866 | 5.887 |
| | ADDDS | **0.257** | **0.256** | **0.252** | **0.262** | **0.263** | **0.254** | **0.270** |
| W2 | ZOD-MC | 6.371 | 10.361 | 10.362 | 10.359 | 10.360 | 10.360 | 10.359 |
| | ADDDS | **4.091** | **4.090** | **4.075** | **4.115** | **4.130** | **4.077** | **4.151** |

*Table C3.* Ablation study for the number of total sampling steps $N$ under $PGD^{10}$ attack.

| | $N$ | 15 | 20 | 25 | 45 | 65 | 85 | 100 |
|---|---|---|---|---|---|---|---|---|
| MMD | ZOD-MC | 5.873 | 6.119 | 5.872 | 5.876 | 5.874 | 5.867 | 5.872 |
| | ADDDS | **0.246** | **0.285** | **0.234** | **0.308** | **0.277** | **0.316** | **0.326** |
| W2 | ZOD-MC | 10.359 | 10.445 | 10.356 | 10.360 | 10.362 | 10.356 | 10.362 |
| | ADDDS | **3.865** | **4.198** | **3.934** | **4.289** | **4.192** | **4.355** | **4.398** |

Figure C1. Results show that ADDDS outperforms other competitors including these two intrinsic defense schemes. Hence the adversarial attacks also threaten these two defense schemes, which strengthens the necessity of developing ADDDS.

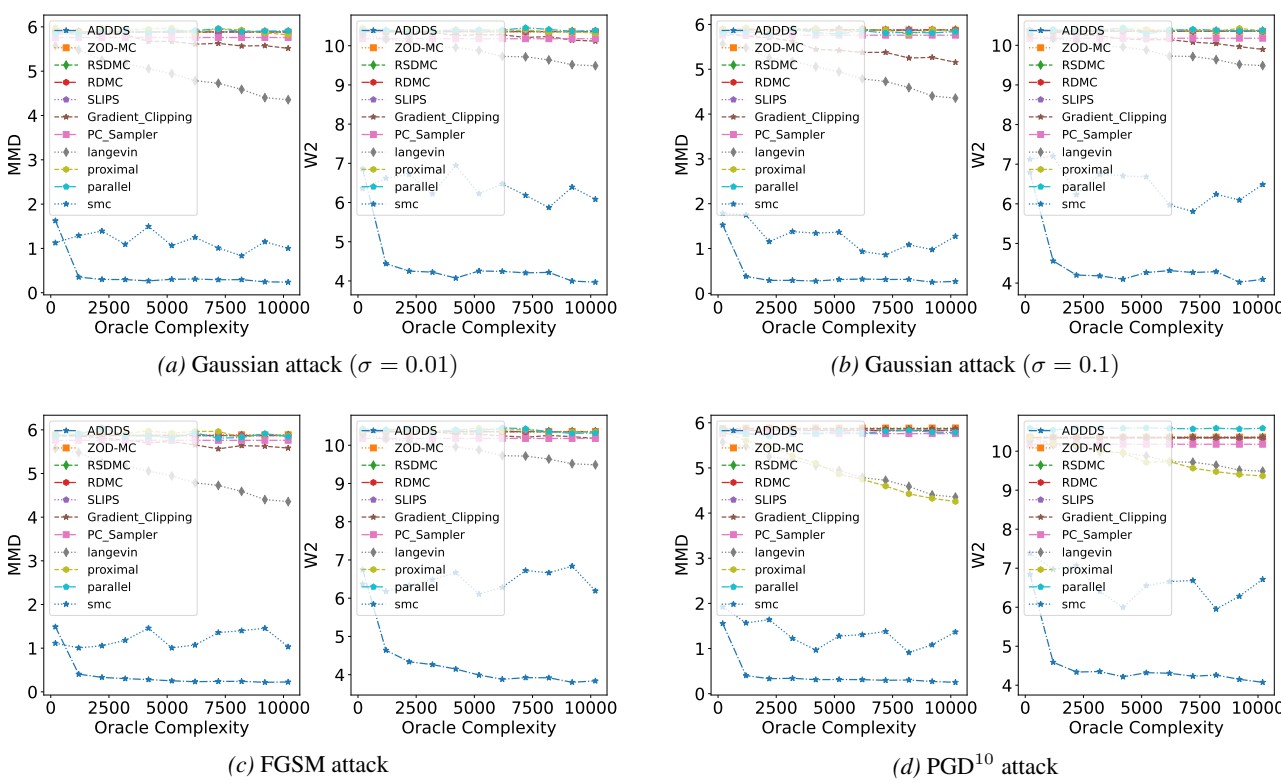

*(a)* Gaussian attack ($\sigma = 0.01$)               *(b)* Gaussian attack ($\sigma = 0.1$)

*(c)* FGSM attack                        *(d)* PGD$^{10}$ attack

*Figure C1.* Sampling accuracy comparison of different methods (including the gradient clipping and the predictor-corrector sampler) for 2D GMM under different attacks.

### C.3. Performance against Adaptive White-box Attack

We also consider a higher threat of the adaptive white-box attack scheme (Tramer et al., 2020), where the attacker knows the exact defense strategy (e.g., Eq. 14 for the case of ADDDS) and further develops a countermeasure to evade the corresponding defense. This adaptive white-box attack is a widely-used and effective method. It already has perfect knowledge of ADDDS and fully exploits the defense mechanism. Figure C2 shows that the defense performance of ADDDS does not deteriorate, whereas other non-defense competitors gain some improvement. Nevertheless, ADDDS is still robust against this adaptive white-box attack.

### C.4. Computational Resource Requirement

To evaluate the performance-efficiency trade-off, we compare the computational resource requirements of different methods on 2D GMM under PGD$^{10}$ attack in Figure C3. Figure C3(a) shows that ADDDS achieves competitive computational time among the compared methods, where the defense optimization takes up less than half of the total time. Figure C3(b) shows that ADDDS achieves competitive memory usage in the low oracle complexity cases ($\leqslant 3200$). From the overall experimental results in this paper, ADDDS already achieves satisfactory recovery performance with a low oracle complexity. Hence ADDDS achieves a good performance-efficiency trade-off. ADDDS requires a little less memory than ZOD-MC because the former just uses the plain (non-strong) Wolfe condition, while the latter uses the strong Wolfe condition.

### C.5. Acceptance Rate and Score Estimation Error

We conduct experiments on the average acceptance rates and score estimation errors of ZOD-MC and ADDDS with 2D GMM under PGD$^{10}$ attack (oracle complexity= 1200). The acceptance rate means the proportion of accepted samples

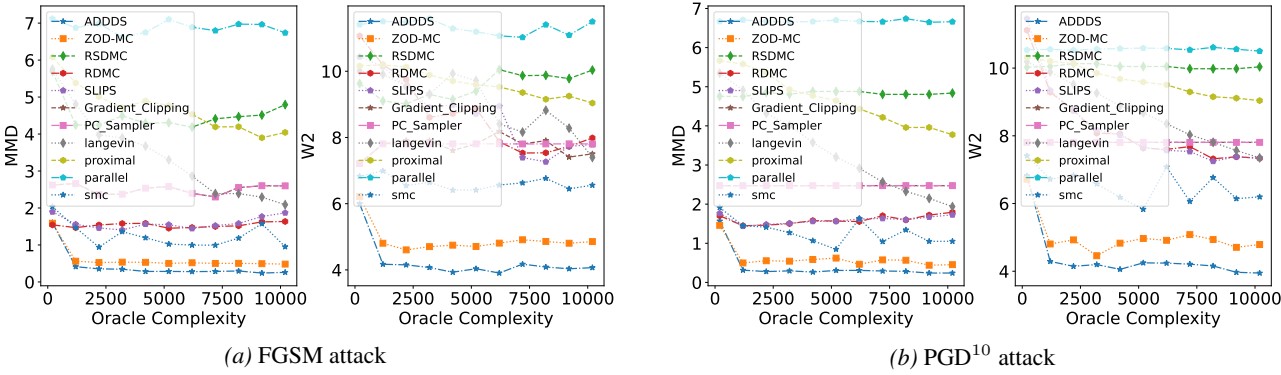

*(a)* FGSM attack

*(b)* PGD$^{10}$ attack

*Figure C2.* Sampling performance comparison on 2D GMM under adaptive white-box attack.

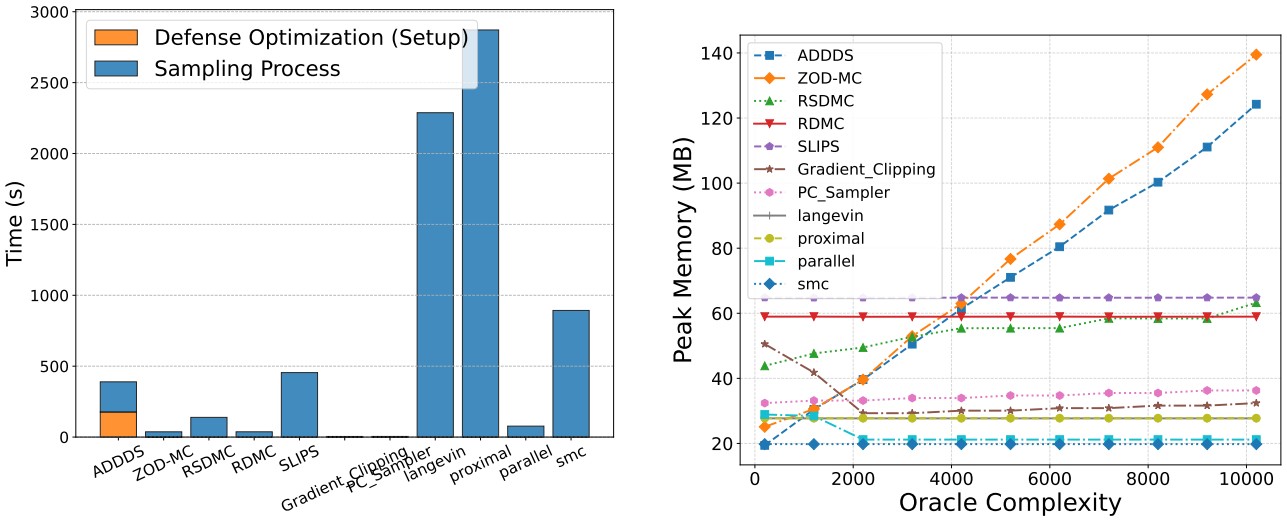

*(a)* Computational Cost Breakdown(Oracle Complexity:10200)

*(b)* Comparison of peak memory usage (MB) across different sampling methods.

*Figure C3.* Computational resource analysis of different methods on 2D GMM under PGD$^{10}$ attack.

$\tilde{z}_{t,i}$ from the rejection sampler, and the score estimation error means $\|s(t, \tilde{x}) - \nabla \log p_t(\tilde{x})\|_2^2$. Both terms are averaged w.r.t. the timestamp $t$. Table C4 shows that ADDDS has a higher acceptance rate than ZOD-MC, which is mainly due to $V(x^\bullet_{ADDDS}) > V(x^\bullet_{org})$. This is consistent with the perturbation tolerance mechanism that adapts to $V(x^* + \delta) > V(x^*)$, as explained in Section 3.2. Moreover, ADDDS achieves a significantly smaller score estimation error than ZOD-MC, which indicates that ADDDS effectively suppresses the perturbation and provides a robust score estimation.

*Table C4.* Average acceptance rate and score estimation error on 2D GMM under PGD$^{10}$ attack.

|  | Acceptance Rate | Score Estimation Error |
|---|---|---|
| ZOD-MC | 2.734% | 14.961 |
| ADDDS | 7.764% | **4.581** |

## C.6. Runtime Comparisons

We provide a direct runtime comparison between ADDDS and first-order DDS methods (SLIPS, RSDMC, and RDMC) for Gaussian mixtures under both PGD$^{10}$ and GCG$^{10}$ attacks (Greedy Coordinate Gradient, Zou et al. 2023) in Table C5. In the early stage, ADDDS needs to solve the LV-regularized minimization problem to find the surrogate minimum $V^\bullet$, which takes up a certain proportion of its total time. However, in the later stage where the surrogate minimum $V^\bullet$ is found and fixed, ADDDS can exploit the zeroth-order rejection sampling scheme, which is much faster than these first-order DDS methods, as indicated by the `Per-iteration Time - Later Stage` and `Single Query Time - Later Stage` columns. Therefore, the computational overhead introduced by LV regularization is controlled within an acceptable scale. Moreover, since ADDDS has the LV regularization defense scheme, it outperforms these first-order baselines in sampling accuracy (see Figures 4(a) and C4). Hence the computational overhead can be compensated by better sampling accuracy.

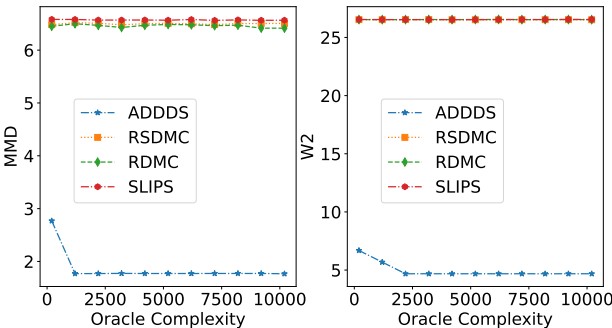

*Figure C4.* Sampling accuracies of different methods w.r.t. oracle complexity for Gaussian mixtures under GCG$^{10}$ attacks. A lower score indicates a better performance.

## C.7. Convergence Curves of Non-smooth Potential Function

To verify that the analyticity assumptions in Appendix A.2 are not critical for the practical performance of ADDDS, we provide empirical convergence curves of the non-smooth potential function $(V(x) + 8\lfloor\|x\|_2\rfloor \mathbb{I}_{\{5<\|x\|_2<11\}})$ for Gaussian mixtures under PGD$^{10}$ attacks in Figure C5. It shows that ADDDS achieves the same convergence performance as that of the zeroth-order method ZOD-MC.

## C.8. Hessian-based Surrogate

We compare the proposed LV regularization with a Hessian-based surrogate (denoted by Hessian-ADDDS) across different sampling tasks, shown in Figures C6, C7, and C8. Results show that Hessian-ADDDS performs much worse than ADDDS, which indicates that the former is not capable of characterizing and suppressing the adversarial perturbations. The Hessian-based surrogate cannot characterize the curvature of an adversarial perturbation, but LV (by its definition) naturally quantifies the variation brought by the adversarial perturbation. Hence ADDDS with LV regularization is preferable to Hessian-ADDDS.

*Table C5.* Runtime comparisons between ADDDS and first-order DDS methods for Gaussian mixtures under PGD[10] and GCG[10] attacks.

**Results under PGD[10] attacks:**

| Dimensionality | Method | Total Time (s) | Per-iteration Time (s) | Single Query Time (ms) |
|---|---|---|---|---|
| 10 | SLIPS | 407.3607 | 18.2672 | 0.001001 |
| 10 | RSDMC | 309.2066 | 14.1894 | 0.001187 |
| 10 | RDMC | 260.0721 | 12.6587 | 0.001042 |
| 10 | ADDDS | 362.5518=Early Stage: 130.4901 +Later Stage: 232.0617 | Early Stage: 14.3729 Later Stage: 8.9145 | Early Stage: 1.6818 Later Stage: 0.000616 |
| 30 | SLIPS | 585.2947 | 25.2828 | 0.001022 |
| 30 | RSDMC | 488.2768 | 20.2605 | 0.001059 |
| 30 | RDMC | 462.1178 | 18.1040 | 0.001021 |
| 30 | ADDDS | 523.5092=Early Stage: 223.8196 +Later Stage: 299.6896 | Early Stage: 21.2554 Later Stage: 12.7525 | Early Stage: 1.6939 Later Stage: 0.000622 |
| 50 | SLIPS | 740.5872 | 40.5750 | 0.001041 |
| 50 | RSDMC | 688.5671 | 36.5524 | 0.000916 |
| 50 | RDMC | 631.3553 | 33.3420 | 0.000947 |
| 50 | ADDDS | 728.6810=Early Stage: 330.0718 +Later Stage: 398.6092 | Early Stage: 38.6613 Later Stage: 23.8593 | Early Stage: 1.7586 Later Stage: 0.000663 |

**Results under GCG[10] attacks:**

| Dimensionality | Method | Total Time (s) | Per-iteration Time (s) | Single Query Time (ms) |
|---|---|---|---|---|
| 10 | SLIPS | 438.3985 | 18.9422 | 0.001040 |
| 10 | RSDMC | 325.8451 | 14.2380 | 0.001007 |
| 10 | RDMC | 289.1137 | 12.8427 | 0.001001 |
| 10 | ADDDS | 375.6102 =Early Stage: 135.3819 +Later Stage: 240.2283 | Early Stage: 14.9386 Later Stage: 9.2471 | Early Stage: 1.7499 Later Stage: 0.000693 |
| 30 | SLIPS | 596.0492 | 25.9084 | 0.001093 |
| 30 | RSDMC | 501.6729 | 20.8636 | 0.001039 |
| 30 | RDMC | 486.3852 | 18.9549 | 0.001035 |
| 30 | ADDDS | 538.7539=Early Stage: 232.4281 +Later Stage: 306.3321 | Early Stage: 21.8460 Later Stage: 13.0429 | Early Stage: 1.7593 Later Stage: 0.000681 |
| 50 | SLIPS | 761.8941 | 41.0285 | 0.001092 |
| 50 | RSDMC | 712.4733 | 37.9204 | 0.001003 |
| 50 | RDMC | 639.8496 | 33.8173 | 0.000995 |
| 50 | ADDDS | 723.3672=Early Stage: 290.7580 +Later Stage: 432.6092 | Early Stage: 38.7448 Later Stage: 24.2957 | Early Stage: 1.7196 Later Stage: 0.000684 |

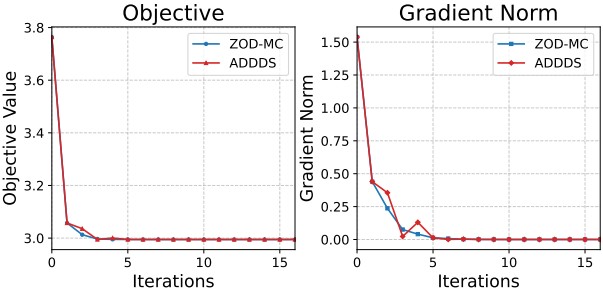

*Figure C5.* Convergence curves of the non-smooth potential function $(V(x) + 8\lfloor \|x\|_2 \rfloor \mathbb{I}_{\{5 < \|x\|_2 < 11\}})$ for Gaussian mixtures under PGD[10] attacks.

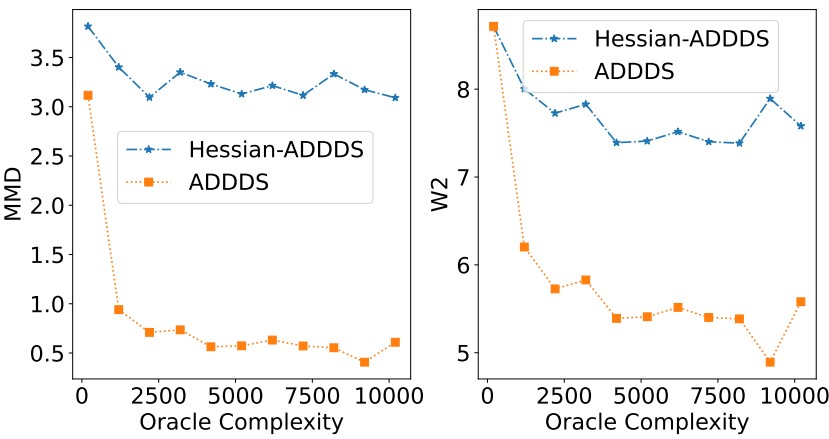

*Figure C6.* Sampling accuracies of different methods w.r.t. oracle complexity for the non-smooth potential function $(V(x) + 8\lfloor \|x\|_2 \rfloor \mathbb{I}_{\{5<\|x\|_2<11\}})$ under $\text{PGD}^{10}$ attacks. A lower score indicates a better performance.

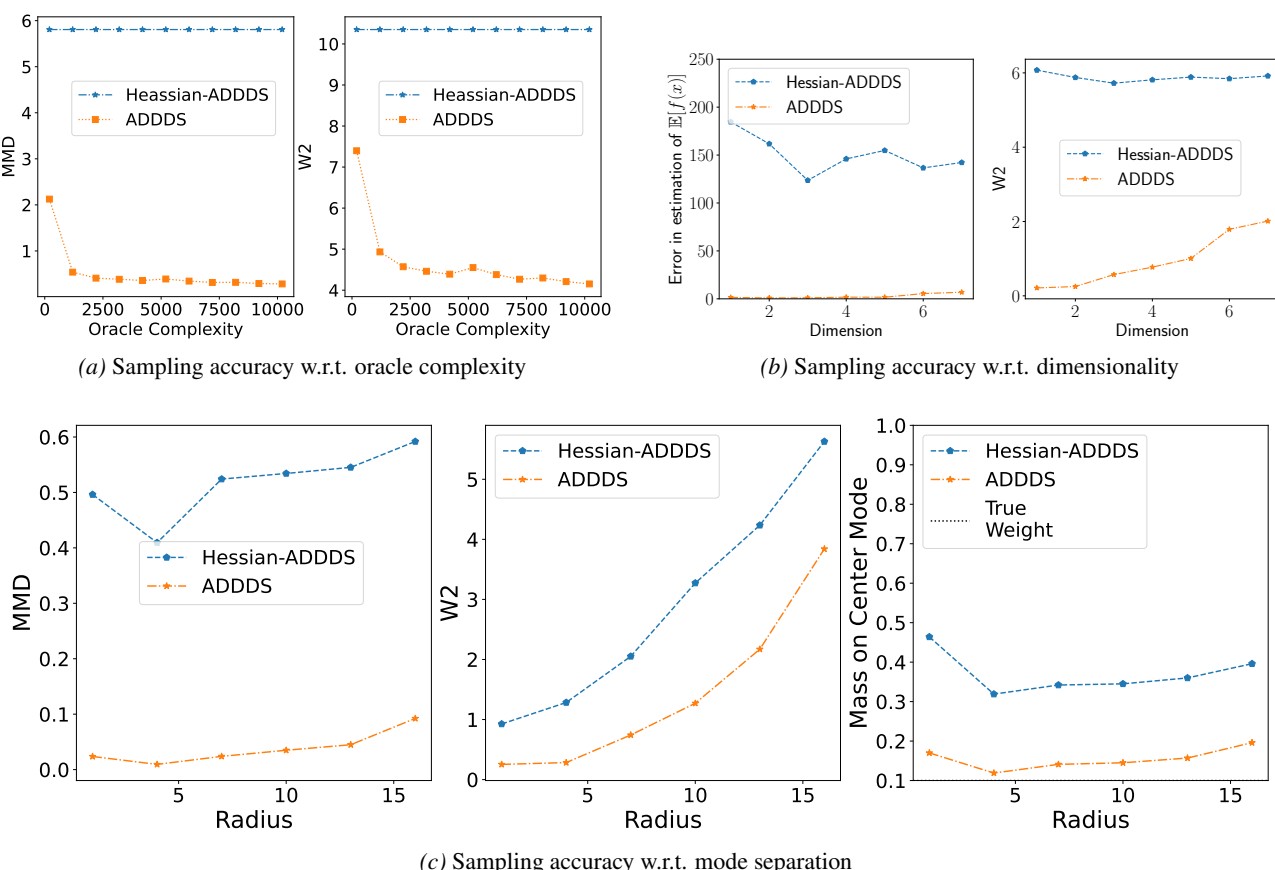

*(a)* Sampling accuracy w.r.t. oracle complexity      *(b)* Sampling accuracy w.r.t. dimensionality

*(c)* Sampling accuracy w.r.t. mode separation

*Figure C7.* Sampling accuracies of Hessian-ADDDS for Gaussian mixtures under $\text{PGD}^{10}$ attacks. A lower score indicates a better performance.

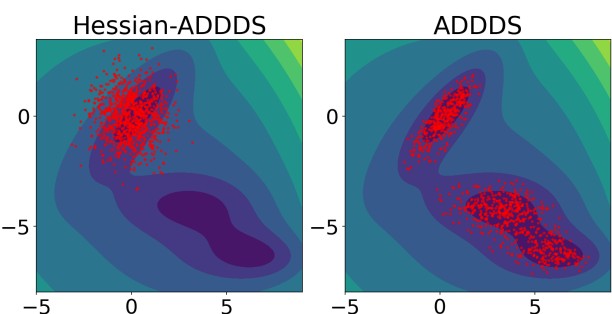

*Figure C8.* Samples generated by Hessian-ADDDS from the Muller-Brown potential under PGD$^{10}$ attacks.

