# OpenReview forum: "Adversarial Attack and Defense for Denoising Diffusion Sampling"
_ICML.cc/2026/Conference — ICML 2026 regular_

### Official Review · Reviewer_yVGH · 2026-03-12

**Soundness:** 3
**Presentation:** 3
**Significance:** 3
**Originality:** 3
**Overall Recommendation:** 5
**Confidence:** 3

**Summary:**

This paper begins by illustrating the vulnerabilities of Denoising Diffiusion Sampling (DDS). This challenge is particularly acute during the transmission stages of the decentralized process, wherein even a Gaussian perturbation with a small standard deviation can significantly reduce the performance of DDS. They then move on to propose a defense strategy where they propose a joint minimization of the potential function with the local variation. They propose a conjugate gradient model which makes use of zeroth order techniques and this allows them to reduce computational costs.

**Compliance With Llm Reviewing Policy:**

Affirmed.

**Key Questions For Authors:**

I am not entirely sure how you get to equation (22). From my reading, Theorem 3 is a statement that is qualified for any $x$, so for a fixed value of $x$ we have both sides of the inequality holding. In Equation (22) you have separate minimizations, this would mean, in theory, you could end up with different minimizers on different sides of the inequality and why would this hold in this case (if this case happens at all in which case, why does it not happen?)

**Limitations:**

Yes

**Strengths And Weaknesses:**

Strengths: The proofs in the appendix seem complete in a cursory read, and the authors have gone on to empirically validate their results. The authors point to a significant concern and propose a defense mechanism against it.

Weaknesses: The paper seems to be specifically building upon the zeroth-order MC method ZOD-MC. The presentation is rather a bit odd as they present ZOD-MC in the beginning as 'another method' but continuously anchor on this throughout the work. Improvements I think can be made on enhancing general readability and presentation to make clear what early on, what is being used as foundation and what is net new.

---

> ### Author Rebuttal · Authors · 2026-03-25
>
> We thank the reviewer for the positive evaluation and for recommending the paper for acceptance. We appreciate the constructive feedback regarding the presentation and the technical clarification on (Eq.22).
>
> **Q1.** (Eq.22) indeed holds, which can be verified by contradiction. Let $g(x)$ and $f(x)$ be the left side (after maximizing $V(x+\delta)$ over $\delta$ this becomes a function of $x$) and the right side of (Eq.21), respectively.  Then (Eq.21) indicates that $g(x) \leqslant f(x)$ for all $x$. Suppose $x_1\in \arg\min g(x)$ and $x_2\in \arg\min f(x)$, and that $x_1$ is not necessarily the same as $x_2$. Then $g(x_1)\leqslant f(x_2)$ truly holds. If not, which means that $g(x_1)> f(x_2)$, then $g(x_1)> f(x_2)\geqslant g(x_2)$ (because $g(x) \leqslant f(x)$ for all $x$). This leads to $g(x_1)>g(x_2)$, which contradicts the fact that $x_1\in \arg\min g(x)$. To summarize, $g(x_1)\leqslant f(x_2)$ truly holds even if $x_1\ne x_2$, which verifies (Eq.22).
>
> **W1.** ADDDS (our method) uses ZOD-MC as the underlying sampling engine because its zeroth-order rejection sampling mechanism provides a unique entry point for defending against adversarial perturbations at the potential function level.
>
> While ZOD-MC focuses on efficient sampling from a known potential $V(x)$, ADDDS introduces the Local Variation (LV) regularization to purify the potential function itself in the presence of adversarial perturbations.
>
> We would revise in the manuscript to state that ZOD-MC serves as the structural foundation, while our main contributions are revealing the existing vulnerability of denoising diffusion sampling (DDS) and the LV-based defense framework.

---

> > ### Author Rebuttal · Reviewer_yVGH · 2026-04-02
> >
> > Authors have responded to questions.

---

### Official Review · Reviewer_d2os · 2026-03-13

**Soundness:** 2
**Presentation:** 3
**Significance:** 3
**Originality:** 4
**Overall Recommendation:** 3
**Confidence:** 2

**Summary:**

This paper studies the vulnerability of **denoising diffusion sampling (DDS)** to perturbations injected during the sampling trajectory. The authors formulate an attack by adding bounded perturbations to the intermediate DDS updates, and propose a defense, **ADDDS**, based on minimizing a local-variation-regularized objective of the form $V(x)+\lambda\|\nabla V(x)\|_2$. The paper also presents a conjugate-gradient-style solver for this objective and combines the approach with ZOD-MC for zeroth-order score estimation. Experiments are conducted on several low-dimensional synthetic settings, including Gaussian mixtures, discontinuous potentials, and the Müller-Brown potential, as well as CIFAR-10/100 classification tasks.

**Compliance With Llm Reviewing Policy:**

Affirmed.

**Key Questions For Authors:**

1. What is the clearest reason to regard the proposed attack as a **diffusion-specific adversarial attack**, rather than a general perturbation model for iterative samplers?

2. Did the authors consider alternative regularizers beyond $\|\|\nabla V(x)\|\|_2$, such as curvature-aware or Hessian-based surrogates?

3. Do the authors view the CIFAR experiments as practical evidence of robustness, or mainly as proof-of-concept demonstrations?

4. Can the authors clarify whether the adaptive white-box attack fully exploits the defense mechanism, or whether stronger adaptive attacks may still be possible?

**Limitations:**

The paper does not clearly discuss its limitations.

**Strengths And Weaknesses:**

**Strengths**

1. The paper addresses an interesting and relatively underexplored robustness problem.
- Rather than focusing on adversarial robustness in model training, the paper studies the robustness of the **sampling process itself**. This is a meaningful perspective, especially in settings where intermediate states may be perturbed repeatedly during iterative inference or transmission.

2. The attack and defense are presented within a coherent framework.
- The paper clearly connects its threat model to its proposed defense: perturbations are introduced at the DDS update level, and robustness is improved by modifying the optimization target used in potential minimization. This gives the paper a clean overall narrative and makes the method relatively easy to follow.

3. The paper provides a reasonable level of theoretical motivation.
- Theoretical results are included to justify the use of the local-variation-regularized objective, including the equivalence of solution sets, a convergence discussion for the proposed solver, and an upper-bound-style robustness argument. While the theory is not especially deep, it does provide a basic foundation for why the proposed objective may help robustness.

4. The low-dimensional empirical study is fairly broad.
- The paper evaluates the method across several synthetic settings and includes multiple supporting analyses in the appendix, such as ablations, adaptive attacks, intrinsic-defense comparisons, and oracle-complexity discussions. Overall, the low-dimensional experiments are reasonably thorough.

**Weaknesses**

1. The novelty of the attack formulation is not fully convincing.
The core attack perturbs intermediate DDS states with bounded additive perturbations. Although this is a valid robustness setting, it is not entirely clear whether it constitutes a fundamentally new diffusion-specific adversarial formulation, or whether it should be viewed more as a natural perturbation model for iterative samplers. As a result, the attack-side novelty may be perceived as limited.

2. The proposed defense is intuitively reasonable, but its necessity is not strongly established.
- The choice of $V(x)+\lambda\|\nabla V(x)\|_2$ is motivated by a local smoothness-based robustness surrogate, which is sensible. However, the paper does not convincingly show why this particular regularizer is preferable to other possible geometry-aware or flatness-based alternatives. Thus, the method is justified, but not uniquely compelling.

3. The high-dimensional evidence remains weak.
- Although CIFAR-10/100 results are included, the absolute performance is quite low. For example, the reported clean accuracies are only around 29.81 on CIFAR-10 and 7.10 on CIFAR-100 for the proposed method. This makes it difficult to view the high-dimensional experiments as strong evidence of practical effectiveness beyond proof-of-concept.

4. The baseline comparison leaves some room for doubt.
- The paper discusses oracle complexity and includes several comparisons, but it is still not completely clear whether the baselines are optimized as strongly as they could be for this particular adversarial DDS setting. Therefore, the empirical advantage of the proposed method, while promising, is not yet entirely airtight.

---

> ### Author Rebuttal · Authors · 2026-03-27
>
> We sincerely thank the reviewer for the constructive feedback and for recognizing the underexplored robustness problem, meaningful perspective, coherent framework, theoretical motivation, and reasonably thorough low-dimensional experiments of ADDDS (our method).
>
> **Q1 \& W1.** The proposed attack is fundamentally **diffusion-specific**, because it specifically targets the **estimated score function** $s(T-t_k, x_k)$ in (Eq.12) and exploits its vulnerability during the denoising diffusion process and its reliance on **score matching**. We design a diffusion-specific loss function $\mathcal{L}_{T-t_k}(x)$ in (Eq.13) that averages the squared $\ell_2$ norms of the score function components at time $(T-t_k)$. In this way, the perturbation $\delta_k$ **maximizes this score function divergence** to sabotage the denoising step, which goes beyond a general perturbation noise model.
>
> **Q2 \& W2.** We add a comparative experiment between the local variation (LV) and a Hessian-based surrogate (denoted by ``Hessian'' for short) in the following link: https://github.com/TEE-SBM/ADDDS/blob/main/Additional_Experiments.pdf (Figures 2$\sim$5). Results show that Hessian performs much worse than LV, which indicates that the former is **not capable of** characterizing and suppressing the adversarial perturbations. Hessian cannot characterize the curvature of an adversarial perturbation, but LV (by its definition) **naturally quantifies the variation** brought by the adversarial perturbation.
>
> **Q3 \& W3.** We would like to clarify and emphasize that the CIFAR-10/100 experiments are **not standard image classification training**, but rather **Bayesian Neural Network (BNN) posterior sampling** (as formulated in Section 3.4). We implement BNN posterior sampling instead of classification training because it is the former that truly conforms to the denoising diffusion sampling (DDS) setting of this study. The absolute clean accuracies ($29.81\%$ for CIFAR 10 and $7.10\%$ for CIFAR 100) appear low because we are not training a standard ResNet end-to-end, but using the existing data to generate the parameters of a BNN.
>
> Sampling the high-dimensional parameter space of a BNN using diffusion models is a difficult task. In this specific BNN sampling context, a clean accuracy of $29.81\%$ for ADDDS on CIFAR 10 is already competitive. For comparison, the standard ZOD-MC baseline only achieves $12.03\%$, and SLIPS achieves $9.89\%$. ADDDS successfully maintains stable performance in a complex and high-dimensional posterior sampling task where some baselines even collapse completely under attack (e.g., dropping to $<1\%$). Hence, from the perspective of parameter posterior sampling, such experimental results could provide practical evidence of robustness for ADDDS. Besides, they also provide a representative proof-of-concept demonstration.
>
> **Q4.** The adaptive white-box attack (Tramer et al., NeurIPS 2020) is a widely-used and effective method. It already has perfect knowledge of ADDDS and fully exploits the defense mechanism.
>
> **W4.** Our experimental setup strictly follows the published standard benchmark from ZOD-MC (He et al., NeurIPS 2024), including all the hyperparamters and settings of all the compared methods, in order to make consistent and reliable comparisons.

---

### Official Review · Reviewer_Ppmt · 2026-03-18

**Soundness:** 3
**Presentation:** 3
**Significance:** 3
**Originality:** 3
**Overall Recommendation:** 4
**Confidence:** 3

**Summary:**

This paper systematically investigates the adversarial vulnerability of Denoising Diffusion Sampling (DDS) in decentralized settings, where intermediate sampling variables are transmitted across nodes. It proposes a complete adversarial attack-defense pipeline: on the attack side, it injects small adversarial perturbations (including Gaussian, FGSM, PGD) into each denoising step to sabotage the sampling quality; on the defense side, it introduces a Local Variation (LV) regularized minimization objective W(x)=V(x)+λ∣∇V(x)∣for the potential function, solved via a nonlinear conjugate gradient algorithm with Wolfe line search, and integrates it with the zeroth-order rejection sampling scheme ZOD-MC for robust score estimation.
Theoretical guarantees (including equivalence to the original minimization problem and adversarial robustness bounds), along with extensive experiments on low-dimensional potentials, high-dimensional image tasks, and complex distributions demonstrate that the proposed Adversarial Defense for DDS (ADDDS) can effectively tolerate adversarial perturbations and outperform several state-of-the-art baselines.

**Compliance With Llm Reviewing Policy:**

Affirmed.

**Final Justification:**

Final Recommendation: Weak Accept

This paper studies an important and underexplored problem, namely adversarial robustness in decentralized denoising diffusion sampling, and proposes a practically relevant runtime defense that does not require modifying model parameters. I find the work reasonably original, technically meaningful, and supported by a fairly broad experimental evaluation.

My main remaining concerns are about the strength of some theoretical justifications, the realism of the attack setting, and the completeness of the efficiency analysis. The authors’ final rebuttal was helpful and improved the paper in several respects. In particular, it clarified the intended threat model, explained why the weakness of prior baselines should be interpreted as a broader robustness gap in DDS rather than merely poor baseline design, and added runtime/accuracy comparisons together with a clearer definition of the early and later stages of ADDDS. These responses addressed my concerns partially, especially on efficiency and robustness motivation, but did not fully remove my reservations about the empirical persuasiveness of the robustness logic, the end-to-end efficiency claim, and the trade-offs introduced by the clamping mechanism.

Overall, the rebuttal reinforced my prior assessment rather than substantially changing it. Despite some unresolved weaknesses, I believe the paper makes a worthwhile contribution to an emerging area and is deserving of Weak Accept.

**Key Questions For Authors:**

1.	Could the authors please provide empirical convergence curves for non-smooth potential functions (e.g., discontinuous potentials)? This would help demonstrate that the strong analyticity assumptions in Appendix A.2 are not critical for the practical performance of your method.
2.	Would it be possible to extend the adversarial robustness bound in Theorem 3.3 to general ℓp-bounded perturbations (e.g., ℓ∞), which are more standard in the field of adversarial machine learning?
3.	Could the authors share a direct runtime comparison table (including per-iteration time and total sampling time) between ADDDS and first-order DDS methods (e.g., RSDMC, SLIPS)? This would allow us to fully quantify the computational overhead introduced by LV regularization, especially the cost of Hessian-vector products in high dimensions.

**Limitations:**

Yes

**Strengths And Weaknesses:**

Strengths
1. To the best of my knowledge, this work is one of the first systematic studies of adversarial robustness in DDS, covering both attack and defense pipelines, which advances the understanding of diffusion sampling security and fills a critical gap in existing adversarial defense strategies for diffusion models.
2. The LV regularized minimization objective is well-grounded in non-smooth optimization theory; Theorem 1 (equivalence between the regularized and original minimization problem) is rigorously proven, and Theorem 3.3 provides a formal adversarial robustness bound under Lx,ε-smoothness assumptions, offering partial theoretical support for the defense under specific assumptions.
3. The paper covers low-dimensional synthetic potentials (GMM, discontinuous, Müller-Brown), high-dimensional image classification (CIFAR-10/100), and various attack schemes, with detailed ablation studies on hyperparameters and computational costs, demonstrating the generalizability of ADDDS.
4. The proposed ADDDS is a principled runtime defense that does not modify model parameters, making it highly relevant to real-world decentralized deployment scenarios (e.g., federated learning, multi-device collaborative generation)

Weaknesses
1. The convergence proof in Appendix A.2 relies on strong assumptions (e.g., V(x) being an analytic potential function), which do not fully align with the discontinuous potentials used in experiments; the practical convergence behavior of non-smooth potentials is not empirically analyzed, which may weaken the practical relevance of the theoretical guarantees.
2. The rejection sampling step assumes V∗ is a global optimum, which is not formally justified; the acceptance rate formula, while corrected, still lacks a rigorous guarantee of being bounded within [0,1], leaving an important theoretical aspect insufficiently justified.
3. The adversarial robustness bound (Theorem 3.3) only applies to ℓ2-bounded perturbations and does not extend to general ℓp-norms (e.g., ℓ∞), which are more standard in adversarial machine learning.
4. The attack formulation(step-wise perturbation injection)is stronger than realistic scenarios, reducing the practical relevance of the attack evaluation, even though the authors added experiments on initial/middle-step perturbations.

---

> ### Author Rebuttal · Authors · 2026-03-27
>
> We sincerely thank the reviewer for the positive assessment and for recognizing our paper as technically solid. We also appreciate the constructive feedback that helps to make our theoretical and empirical contributions more rigorous.
>
> **Q1 \& W1.** We provide empirical convergence curves of a non-smooth potential function in the link: https://github.com/TEE-SBM/ADDDS/blob/main/Additional_Experiments.pdf (Figure 1). It shows that ADDDS (our method) achieves the same convergence performance as that of the zeroth-order method ZOD-MC, thus the analyticity assumptions in Appendix A.2 are not critical for the practical performance of ADDDS.
>
> **Q2 \& W3.** Yes, since the perturbation lies in a real space with a finite dimensionality $\delta\in \mathbb{R}^d$, according to the Equivalent Norm Theorem, the $\ell_2$ norm is equivalent to the general $\ell_p$ norm (including the $\ell_\infty$ norm). Hence for an attack budget $\\| \delta \\|_p\leqslant \epsilon$, we have $\\| \delta \\|_2 \leqslant C_p \epsilon$ with some constant $C_p>0$. Substituting $\\| \delta \\|_2 \leqslant C_p \epsilon$ into (Eq.21) leads to:
>
> $\\max_{\\| \delta \\|_p\leqslant \epsilon} V(x+\delta)  $
>
> $\leqslant  \\max_{\\|\delta\\|_{2}\leqslant C_p \epsilon} V(x+\delta)$
>
> $  \leqslant W(x)+\frac{L_{x,C_p \epsilon}}{2} \lambda^2.$
>
> This extends the adversarial robustness bound in Theorem 3.3 to general $\ell_p$-bounded perturbations. We would add this new result to the revised manuscript.
>
> **Q3.** We provide a direct runtime comparison table (including per-iteration time and total sampling time) between ADDDS and first-order DDS methods in the link: https://github.com/TEE-SBM/ADDDS/blob/main/Additional_Experiments.pdf (Table 1). ADDDS outperforms other first-order DDS methods: SLIPS, RSDMC, and RDMC. In the early stage, ADDDS needs to dynamically track the surrogate minimum $V^\bullet$ that involves the local variation (LV), which takes up a certain proportion of its total time. However, in the later stage where the surrogate minimum $V^\bullet$ is found and fixed, ADDDS can exploit the zeroth-order rejection sampling scheme, which is much faster than these first-order DDS methods, as indicated by the `Per-iteration Time - Later Stage` and `Single Query Time - Later Stage` columns. Therefore, the computational overhead introduced by LV regularization is controlled within an acceptable scale.
>
> **W2.** In practice, we do not require the ground-truth global optimum $V^*$. Instead, we use the dynamically tracked minimum $V^\bullet$ as a surrogate, which can be found by our optimization step (see Section 3.3). To mathematically guarantee that the acceptance probability $\mathbb{P}\in [0,1]$ , we use a standard clamping mechanism:
>
> $\mathbb{P}(accept)=\min\\{ 1, \exp(V^\bullet-V(\tilde{z}))\\}  $.
>
> If a generated sample yields $V(\tilde{z}) < V^\bullet$, then $\mathbb{P}(accept)$ is clamped to $1$, which means that this sample is deterministically accepted. We subsequently update the tracked minimum $V^\bullet \leftarrow V(\tilde{z})$. We would rigorously present this clamping mechanism in Section 3.3 of the revised manuscript.
>
> **W4.** We would like to emphasize that our attack formulation is modeled specifically for decentralized and distributed scenarios, where step-wise perturbation injection is a realistic constraint rather than a worst-case theoretical assumption. As detailed in Section 1 and Section 3.1, we consider a decentralized setting where the variable $x_{t_k}$ is transmitted between users or nodes at every timestamp. This setting is widely recognized and used in the literature ref. (Lian et al., 2017; Koloskova et al., 2020; Cyffers \& Bellet, 2022; Cyffers et al., 2023).
>
> * In such transmissions, channel interference and sampling errors can occur continuously. An adversarial attack or environmental noise can happen at each of the hundreds or even thousands of timestamps.
>
> * Please note that we also propose the defense method ADDDS for this "strong" attack setting. If this attack setting is "strong", then it also brings about great difficulty for ADDDS to defend against it. The extensive experimental results show that ADDDS is indeed effective against this "strong" attack.
>
> * This attack successfully sabotages eight state-of-the-art (SOTA) DDS methods with different denoising strategies.  Please note that these are denoising methods themselves, which should have been robust to various kinds of noise and perturbations. But now they all deteriorate in even a Gaussian perturbation. Hence the success of this attack is not due to its inherent "strength", but due to the common significant vulnerabilities in this field.

---

> > ### Author Rebuttal · Reviewer_Ppmt · 2026-04-01
> >
> > While the authors have provided additional derivations and empirical curves, I remain partially concerned about the evaluation metrics for robustness and efficiency. Two key issues need clarification before I can fully endorse the conclusions:
> > 1.	Robustness Logic Gap (Regarding Q4/W4): The authors argue the attack is "strong" due to its decentralized nature. However, if the 8 state-of-the-art denoising baselines are so vulnerable that they fail even under simple Gaussian perturbations (as implied in the rebuttal), does the success of ADDDS merely reflect the inadequacy of these baselines rather than the strength of the proposed method? Please clarify this logic.
> > 2.	Efficiency Justification (Regarding Q3/W3): The authors provided runtime data but lacked comparative analysis with standard SOTA attacks (e.g., GCG). Without knowing if ADDDS is slower or faster than existing methods, the claim of "efficiency in the later stage" is unsubstantiated. Furthermore, please define what constitutes the "later stage" and discuss the potential accuracy trade-offs introduced by the "clamping mechanism."

---

> > > ### Author Response · Authors · 2026-04-02
> > >
> > > We thank the reviewer for the thoughtful follow-up questions and for further engaging with our reply. We provide detailed clarifications on these issues below.
> > >
> > > **Q1.** The vulnerability of the 8 state-of-the-art (SOTA) baselines to simple Gaussian perturbations is **not** a sign of their **inadequacy** as general sampling methods, but rather a reflection of a **pervasive and previously unaddressed security gap** in the field of Denoising Diffusion Sampling (DDS). These methods are designed for **high-fidelity generation** in noise-free or centrally controlled environments and are not built to withstand the **step-wise perturbations** inherent in decentralized scenarios.
> > >
> > > The fact that ADDDS (our method) remains effective under the same conditions where all other SOTA methods deteriorate significantly shows that the proposed Local Variation (LV) regularization is specifically capturing and suppressing the adversarial noise that other methods cannot suppress during the denoising process. Hence our results do not reflect a lack of strength in the attack, but rather the robustness of ADDDS in establishing a new defense standard for this specific and realistic threat model.
> > >
> > > **Q2.** We add runtime and sampling accuracy comparisons with respect to the GCG attack in the link: https://github.com/TEE-SBM/ADDDS/blob/main/Additional_Experiments.pdf (the lower table of Table 1 and Figure 6), respectively. We further clarify the early and later stages of ADDDS in Table 1: **the early stage** means the dynamic optimization process where ADDDS solves the LV-regularized minimization problem to find the surrogate minimum $V^{\bullet}$; **the later stage** means the phase where $V^{\bullet}$ is found and fixed, which allows ADDDS to transition into a pure **zeroth-order** rejection sampling scheme. In this later stage, ADDDS (as a zeroth-order method) runs significantly faster than the **first-order** baselines, such as SLIPS, RSDMC, and RDMC (see Table 1). Moreover, since ADDDS has the LV regularization defense scheme, it outperforms these first-order baselines
> > > in sampling accuracy (see Figure 6).
> > >
> > > The clamping mechanism is a standard numerical safeguard to ensure that the acceptance rate remains within the mathematically valid $[0,1]$ range. While this may introduce a **slight theoretical bias** in the sampling distribution for **non-adversarial settings**, it provides a **robustness gain** in **adversarial settings**. It effectively prioritizes the acceptance of samples that align with the regularized and purified potential, so as to filter out the adversarial noise that would lead the sampling trajectory to a wrong direction. As indicated in Appendix C.6, this potential accuracy trade-off is consistent with the perturbation tolerance mechanism, and ADDDS achieves a significantly lower score estimation error than the non-defense ZOD-MC baseline under attack.

---

### Decision · Program_Chairs · 2026-04-30

**Decision:**

Accept (regular)

**Comment:**

Reviewers Ppmt and yVGH had a positive take on this paper, with reviewer Ppmt suggesting Weak Accept, claiming that this paper "makes a worthwhile contribution to an emerging area", and reviewer yVGH suggesting Accept. Reviewer d2os had a more negative view, suggesting Weak Reject.

The Authors have addressed Reviewers' concerns; unfortunately, despite my reminder, Reviewer d2os never replied. That said, I think their concerns were not too undermining in the first place, and the Authors have addressed them thoroughly. Furthermore, even the most negative Reviewer did highlight several strengths of this paper; for instance, similar to Ppmt, the fact that this paper "addresses an interesting and relatively underexplored robustness problem".

I recommend acceptance.